# Cortico-striatal action control inherent of opponent cognitive-motivational styles

Cassandra Avila[1], Martin Sarter[1,2]*

[1]Department of Psychology, University of Michigan, Ann Arbor, United States; [2]Department of Psychology & Neuroscience Program, University of Michigan, Ann Arbor, United States

## eLife Assessment

This **valuable** manuscript investigated the role of glutamate signaling in the dorsomedial striatum of rats in a treadmill-based task and reported that it differs in goal-trackers compared to sign-trackers in a way that corresponds to differences in behaviour. The evidence supporting these claims is **solid** but could be further strengthened by adding more analyses and more detailed descriptions of current analyses. These findings will primarily be of interest to behavioural neuroscientists.

*For correspondence:
msarter@umich.edu

**Abstract** Turning on cue or stopping at a red light requires attending to such cues to select action sequences, or suppress action, in accordance with learned cue-associated action rules. Cortico-striatal projections are an essential part of the brain's attention–motor interface. Glutamate-sensing microelectrode arrays were used to measure glutamate transients in the dorsomedial striatum (DMS) of male and female rats walking a treadmill and executing cued turns and stops. Prelimbic–DMS projections were chemogenetically inhibited to determine their behavioral necessity and the cortico-striatal origin of cue-evoked glutamate transients. Furthermore, we investigated rats exhibiting preferably goal-directed (goal trackers, GTs) versus cue-driven attention (sign-trackers, STs), to determine the impact of such cognitive-motivational biases on cortico-striatal control. GTs executed more cued turns and initiated such turns more slowly than STs. During turns, but not missed turns or cued stops, cue-evoked glutamate concentrations were higher in GTs than in STs. In STs, turn cue-locked glutamate concentrations frequently peaked twice or three times, contrasting with predominately single peaks in GTs. In GTs, but not STs, inhibition of prelimbic–DMS projections attenuated turn rates and turn cue-evoked glutamate concentrations and increased the number of turn cue-locked glutamate peaks. These findings indicate that turn cue-evoked glutamate release in GTs is tightly controlled by cortico-striatal neuronal activity. In contrast, in STs, glutamate release from DMS glutamatergic terminals may be regulated by other striatal circuitry, preferably mediating cued suppression of action and reward tracking. As cortico-striatal dysfunction has been hypothesized to contribute to a wide range of disorders, including complex movement control deficits in Parkinson's disease and compulsive drug taking, the demonstration of phenotypic contrasts in cortico-striatal control implies the presence of individual vulnerabilities for such disorders.

## Introduction

Cue-triggered selection of action, or suppression of action, are essential components of adaptive behavior. Fronto-striatal projections have been thought to play an essential role in the integration of information about attended cues into striatal circuitry, to prioritize cue-linked action and facilitate modification of action selection in response to changing action outcomes (*Balleine and O'Doherty, 2010*; *van Schouwenburg et al., 2012*; *Chatham et al., 2014*; *Hart et al., 2018a*; *Hart et al., 2018b*).

Deficient or biased cortico-striatal functions has been postulated to cause neuropsychiatric symptoms ranging from complex movement control deficits in Parkinson's disease (PD) to compulsive addictive drug use (*Volkow et al., 2006*; *Bohnen et al., 2009*; *Ersche et al., 2011*; *Marshall and Ostlund, 2018*; *Rasooli et al., 2021*; *Sarter et al., 2021*).

The cortical control of striatal action selection has been extensively supported by studies investigating the effects of manipulation of the excitability of fronto-striatal neurons in non-human animals (for review see *Sharpe et al., 2019*) and fronto-cortical functional connectivity in humans (*Postuma and Dagher, 2006*; *Shepherd, 2013*; *Devignes et al., 2022*). The present experiments were designed to directly determine the fronto-striatal representation of action-initiating and action-suppressing movement cues. Focusing on the prelimbic projections to dorsomedial striatum (DMS) (*Mailly et al., 2013*), that previously were demonstrated to be necessary for learning goal-directed actions (*Hart et al., 2018b*; *Choi et al., 2023*), we measured real-time, cue- and reward-locked glutamate concentrations in the DMS using amperometry and glutamate-sensing electrodes. Furthermore, we inhibited prelimbic–DMS projections to attribute such release to activity in this pathway and assess the necessity of cortico-striatal glutamate release for cue-evoked movement and suppression of movement.

Glutamate transients were recorded in rats performing visual and auditory cue-triggered turns and stops while walking a treadmill. Following a turn cue, the treadmill stopped and restarted in reverse direction while, following a stop cue, it restarted in the same direction. The development of this Cued-Triggered Turning Task (CTTT; *Avila et al., 2020*) was inspired by evidence showing that people with PD who experience a propensity for falls exhibit deficient turns (*Stack and Ashburn, 2008*; *Cheng et al., 2014*). In people with PD, such an elevated risk for falls has been attributed to loss of cholinergic neurons innervating cortical regions and the resulting failure to attend to, and thus select and evaluate, movement cues (*Bohnen et al., 2009*; *Yarnall et al., 2011*; *Rochester et al., 2012*; *Bohnen et al., 2019*; *Kim et al., 2019*). In conjunction with the disease-defining striatal dopamine losses, degraded information about movement cues in the striatum is thought to further slow and disrupt the cue-guided selection and sequencing of movements. Moreover, in interaction with striatal dopamine losses, impaired cortico-striatal processing of movement cues would be expected to enhance the detrimental impact of attentional distractors on movement sequences, as indicated by the capacity of such distractors to evoke freezing of gait (e.g., *Cowie et al., 2012*) and movement errors and falls (*Kucinski et al., 2013*; *Sarter et al., 2014*; *Albin et al., 2021*; *Sarter et al., 2021*; *Albin et al., 2022*). We previously demonstrated the usefulness of the CTTT for modeling clinically relevant, attentional-motor deficiencies by showing that rats with combined cortical cholinergic and striatal dopaminergic losses and a propensity for falls (*Kucinski et al., 2013*; *Kucinski et al., 2019*; *Kucinski and Sarter, 2021*; *Sarter et al., 2021*) exhibit deficits in cued turning, but not cued stopping (*Avila et al., 2020*).

The present experiments also sought to demonstrate that variations in the individual capacity for voluntary, or top–down, attentional control influence the efficacy of cortico-striatal control of action selection. Successfully performing well-practiced, visual or auditory cued turns and stops requires sustaining attention to cue sources, the continued selection of cues as signified by the behavior evoked by cues, monitoring the state of the treadmill and action outcome, and maintaining cue modality-linked response rules in working memory. To reveal the impact of variations in such attentional capacities, we investigated rats that exhibit, as traits, opponent attentional biases. These rats are selected from outbred populations using a Pavlovian Conditioned Approach (PCA) test. Goal-trackers (GTs) orient to a Pavlovian reward cue, that is, they learn its predictive significance, but they do not approach and contact such a cue. In contrast, sign-trackers (STs) approach and contact such a cue, which has been interpreted as assigning incentive salience to a Pavlovian cue and attributed to cue-evoked mesolimbic dopamine signaling (*Flagel et al., 2011*; *Flagel and Robinson, 2017*; *Iglesias et al., 2023*). The relatively greater vulnerability of STs for addictive drug taking and relapse in the presence of Pavlovian drug cues has been extensively documented (e.g., *Saunders and Robinson, 2010*; *Yager and Robinson, 2010*; *Yager and Robinson, 2013*).

Goal- and sign-tracking index the presence of broader, opponent cognitive-motivational styles that are dominated by a bias toward deploying top–down (or goal-directed) and bottom–up (or stimulus-driven) processing of action cues, respectively (for review see *Sarter and Phillips, 2018*; for evidence of the presence of these traits in humans see *Schad et al., 2020*; *Colaizzi et al., 2023*). As such biases of GTs and STs were previously shown to be mediated in part via contrasting cholinergic capacities for the detection of cues (*Paolone et al., 2013*; *Koshy Cherian et al., 2017*; *Pitchers et al., 2017a*;

*Pitchers et al., 2017b*), we hypothesized that contrasts in the cortico-striatal processing of movement cues contribute to the expression of these opponent biases. The main results from the present experiments indicate that GTs utilize turn cues more efficiently than STs and that in GTs, but not STs, cued turning is tightly controlled by prelimbic–DMS glutamate signaling.

## Results
### Phenotype screening and distribution by sex and vendor
PCA-based screening generated five behavioral measures indicating the speed and frequency of contacting the lever (Pavlovian conditioned stimulus [CS]) versus the speed and frequency of head entries into the food port. These measures were collapsed into the PCA score, indicating the degree of sign- and goal-tracking, respectively, of individual rats over five PCA test sessions. Prior analyses have consistently shown that all rats orient toward the CS, that is, all rats learn the predictive significance of the CS, but only some – the STs – approach and contact the CS (*Figure 1a*, top right photo), while GTs do not contact the CS but approach and enter the food port (*Figure 1a*, bottom right photo; e.g., *Robinson and Flagel, 2009*; *Meyer et al., 2012*; *Pitchers et al., 2017a*).

PCA scores from the last two sessions were averaged and used to classify rats as STs and GTs. PCA screening of $N = 378$ (215 females) rats yielded 113 GTs (30%), 155 rats with intermediate scores (INs; 41%), and 110 STs (29%). *Figure 1a* shows the distribution of PCA scores across the five test sessions of the rats used for subsequent CTTT training and amperometric recordings (see *Figure 2* for the number of rats used for individual experiments; INs were not used in these experiments and *Figure 1a* does not depict their PCA scores).

As PCA screening was conducted in four separate cohorts of rats, separated by 2–19 months, we first determined that PCA scores did not differ across these cohorts (no main effects of cohort on response bias (respbias), probability difference (probdiff), latency, or PCA index; all $F < 1.74$, all $p > 0.16$). Chi-square tests were used to determine if sex or the commercial source of the rats (vendor) influenced the distribution of phenotypes. Consistent with prior reports (*Pitchers et al., 2015*), the distribution of PCA scores did not differ by sex ($X^2(2, N = 378) = 0.35$, $p = 0.56$; *Figure 1b*). However, the distribution of PCA scores from rats obtained from the two vendors differed significantly ($X^2(2, N = 378) = 36.62$, $p < 0.001$; *Figure 1c*; note that the percentages of phenotypes shown in *Figure 1c* were calculated individually for each vendor). Relatively more rats obtained from Inotiv displayed lever-directed behaviors, while rats supplied by Taconic relatively more frequently exhibited goal cup-directed behaviors (see also *Fitzpatrick et al., 2013*; *Gileta et al., 2022*). A follow-up analysis rejected the possibility that vendor-specific PCA score distribution differed by sex (Inotiv: $X^2(2, N = 133) = 1.30$, $p = 0.52$; Taconic: $X^2(2, N = 245) = 1.41$, $p = 0.50$). GTs and STs of both sexes and from either vendor were combined for use in subsequent experiments.

### CTTT acquisition and criterion performance
Upon reaching criterion performance in the CTTT (see the apparatus, illustration of task rules and a timeline of an individual trial in *Figure 3*), defined as >70% cued turns and cued stops for two consecutive days/sessions, and prior to the intracranial implantation of microelectrode arrays (MEAs), the relative number of cued turns and stops were recorded from an additional four test sessions to determine baseline CTTT performance of GTs and STs.

The number of training sessions required by GTs ($n = 29$, 13 females) and STs ($n = 22$, 12 females; *Figure 2*) to reach CTTT criterion performance, defined as 70% correct responses to either cue for two consecutive sessions, did not differ significantly ($F(1,47) = 0.01$, $p = 0.94$; *Figure 4a*). Likewise, male and female rats acquired this task at comparable rates (main effect of sex: $F(1,47) = 0.81$, $p = 0.37$; phenotype × sex: $F(1,47) = 0.27$, $p = 0.61$).

After having reached performance criterion, GTs scored significantly more cued turns than STs across four subsequent test sessions (main effect of phenotype: $F(1,47) = 5.03$, $p = 0.003$, $\eta_p^2 = 0.097$; *Figure 4b*). This analysis also revealed a main effect of day (or session; $F(3,141) = 4.96$, $p = 0.003$, $\eta_p^2 = 0.096$), reflecting that all rats generated more cued turns during the third when compared with the first of the four sessions used for the determination of CTTT baseline performance (for multiple comparisons see *Figure 4c*). Sex did not affect turning rates ($F(1,47) = 0.01$, $p = 0.95$) and interactions between phenotype, sex, and day remained insignificant (all $F < 2.10$, all $p > 0.15$). The proportion of

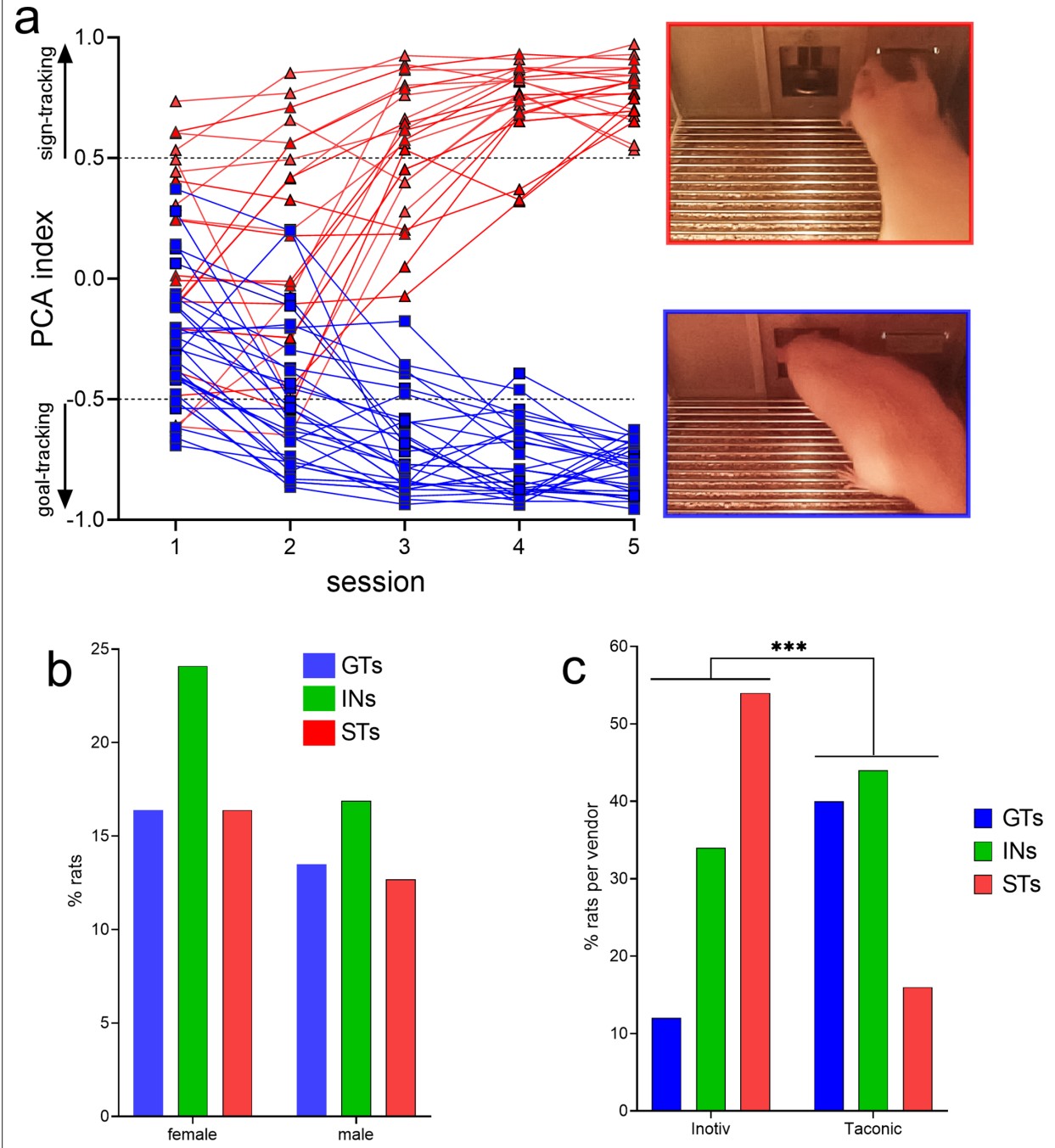

**Figure 1.** Results of behavioral phenotyping. (**a**) Distribution of Pavlovian Conditioned Approach (PCA) scores of rats eventually categorized as goal trackers (GTs) or sign-trackers (STs). A total of $N = 378$ rats (215 females) underwent PCA screening. Final phenotypic classification was based on PCA score averages from the fourth and fifth test session (rats with intermediate PCA scores, INs, are not shown). The photos on the right depict lever contact triggered by the conditioned stimulus (CS) and indicative of sign-tracking behavior (top), versus and CS-triggered food port entry, signifying goal-tracking behavior (bottom). The distribution of PCA scores was unaffected by sex (**b**) but differed among rats obtained from the two vendors (**c**; ***, p<0.001). Relatively more rats from Inotiv were classified as STs while rats from Taconic tended to emerge relatively more often as GTs (note that in <u>c</u> phenotype percentages are expressed based on the total number of rats per vendor). The distribution of vendor-specific PCA scores was not affected by sex (see Results; not shown).

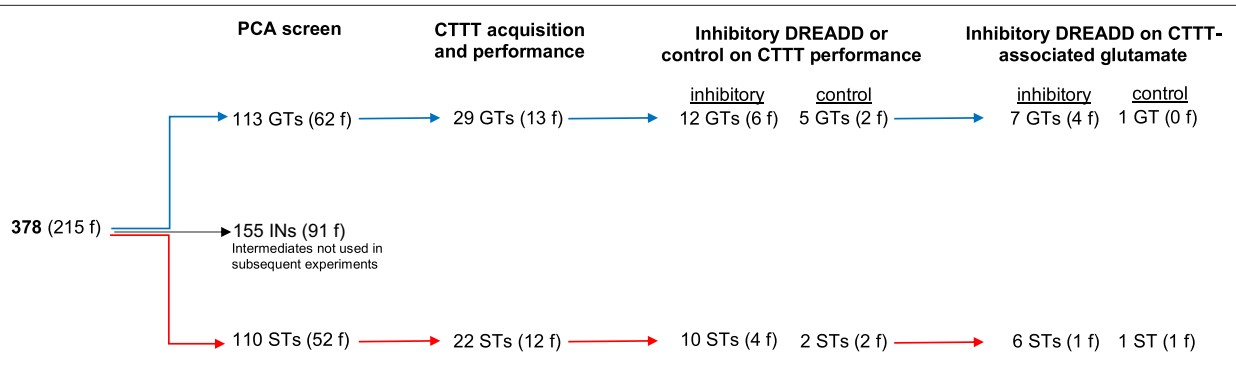

**Figure 2.** Number of rats undergoing Pavlovian Conditioned Approach (PCA) screening, Cued-Triggered Turning Task (CTTT) acquisition and performance testing, and amperometric recordings (f, females).

cued stops following stop cue presentation did not differ by phenotype, day, or sex (all main effects and interactions: all $F < 3.06$; all p > 0.052; *Figure 4d*).

Relative to the time of cue onset, GTs initiated and completed cued turns significantly later than STs (main effects of phenotype; initiation: $F_{(1,51)} = 8.84$, p = 0.005, $\eta_p^2 = 0.16$; completion: $F_{(1,51)} = 9.40$, p = 0.004, $\eta_p^2 = 0.17$; *Figure 4e*; effects of sex and interactions between the effects of sex and phenotype: all $F < 1.03$, all p > 0.32). Reflecting the parallel effects of phenotype on initiation and completion time relative to cue onset, the time needed to complete turns, that is, the time from turn initiation to completion, did not differ between the phenotypes (main effects of phenotype, sex, and interactions: all $F < 3.51$, all p > 0.07; GTs: 2.7 ± 0.2 s (M, SEM); STs: 2.3 ± 0.2 s).

Taken together, GTs and STs acquired the CTTT within a similar number of training sessions, GTs scored more cued turns than STs, and GTs initiated and completed cued turns, relative to cue onset, at significantly later times relative to cue onset. Subsequent experiments recorded DMS glutamatergic transients during cued turns, cued stops, missed turns, false turns, locked to reward delivery, and tested the role of fronto-striatal glutamatergic projections in generating performance-associated glutamatergic transients. Because of the absence of significant effects of sex, and of interactions involving sex as a factor, in the analysis of baseline CTTT performance, subsequent statistical analyses, using linear-mixed effect models (LMMs) to account for effects of animals and variable numbers of transients recorded from each animal, did not further involve sex as a statistical variable. Furthermore, cue modality did not impact turn or stop rates and thus was not included as a factor in the final analyses.

## Turn and stop cue-locked, phenotype-specific glutamate dynamics

Rats were screened for the classification of the phenotypes, underwent CTTT training until they performed at criterion level, followed by implantation of an MEA into the DMS and subsequent recordings of glutamate currents during CTTT performance. *Figure 5* illustrates the four recording sites fabricated onto a ceramic backbone (*Figure 5a*), the preparation of pairs of recording sites for the measurement of glutamate concentrations and potential electroactive interferents, and the measurement scheme (*Figure 5b, c*). Following calibration of glutamate oxidase-coated and sentinel electrodes in vitro (*Table 1*; a representative example is shown in *Figure 5d*), MEAs were permanently implanted into the DMS (*Figure 5e*; see Methods for stereotaxic coordinates of accepted placements).

*Figure 5a, b* shows representative glutamate currents recorded during cued turns in GTs and STs. These traces illustrate the predominance of single, turn cue-locked peaks in GTs, contrasting with the more frequent presence of multiple (two or three) peaks with relatively smaller maximum amplitudes in STs (see Methods for the definition of a peak). Furthermore, reward delivery-locked peaks occurred more frequently, and with relatively higher maximum amplitudes, in STs than in GTs. Our analyses of glutamate peak characteristics were guided by the view that such peaks reflect the orchestrated yet asynchronous depolarization of glutamatergic terminals, over several hundreds of milliseconds and within fractions of micrometers from the electrode surface, and likely sufficient to stimulate synaptic and extra-synaptic, relatively low-affinity glutamate receptors (*Hascup et al., 2008*; *Parikh et al., 2008*; *Parikh et al., 2010*; *Mattinson et al., 2011*; *Quintero et al., 2011*; *Parikh et al., 2014*) (e.g.,

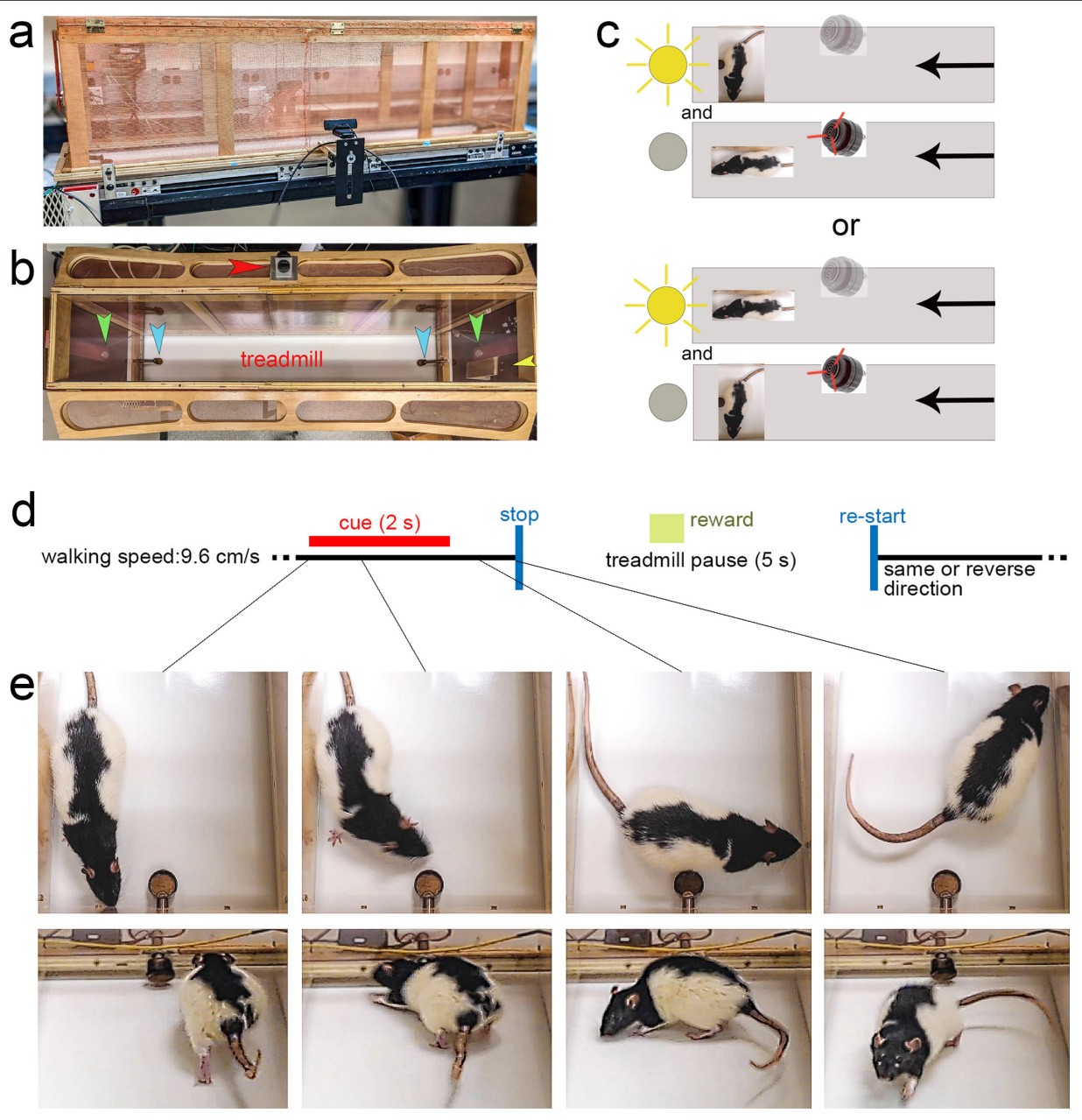

**Figure 3.** Illustration of the Cued-Triggered Turning Task (CTTT), task rules, trial timeline, and of a cued turn. (**a**) shows a lateral view of 1.5 m long CTTT, with Faraday shielding comprised of copper wire mesh surrounding the motor and wooden frame of the Plexiglas-lined enclosure (both grounded), two mounted cue lights on both ends, one mounted SonAlert device centered (not visible here), and a copper reward port installed on either end of the treadmill. Web cameras were placed on each side of the treadmill for offline analysis of the rat's performance. (**b**) Top–down view of the treadmill apparatus showing the position of the stimulus lights (green arrows), the SonAlert device to generate the tone (red arrow), the two copper reward ports (blue arrows), and a camera situated outside of the enclosure (yellow arrow). As illustrated in (**c**) half of the rats were trained to turn in response to the light stimulus and to stop in response to the tone (upper two panels), and the other half acquired the reversed stimulus–response rules lower two panels; note that in this example the treadmill moves from right to left – see arrows on the right – and, in case of turn cue would restart after the pause – see (**d**) – moving from left to right. (**d**) Trial events and timeline. The CTTT required rats to walk on a treadmill (walking speed: 9.6 cm/s). Presentation of a turn or stop cue for 2 s indicated that the treadmill stopped 1 s later, and restarted after a 5-s pause, in the reverse or same direction, respectively. Rats were trained using either the tone as the turn cue and the light as the stop cue, or vice versa. Successful turns and stops without turns, respectively, were rewarded by delivering a 45 mg banana pellet, on average 3.6 s following cue offset (see green square in **d**). Cued turns and cued stops were rewarded at the port located in front of the rat following completion of the turn or the stop, respectively. (**e**) shows photographs of a cued turn, initiated within the first second of cue onset (see lines connecting the individual photos to the task timeline in (**d**); the upper four photographs were extracted from a top–down video and the lower four from a video taken using a camera situated at the backside of the apparatus (yellow arrow in **b**), facing the rat following

*Figure 3 continued on next page*

Figure 3 continued

completion of the turn). Successive presentation of cues was separated by a variable intertrial interval (ITI) of 60 ± 45 s during which the rat continued walking the treadmill. Note that the rat shown in the photographs is a Long-Evans rat that was not part of the present experiments. This rat was used to generate these photos as the black-spotted fur provided better contrast against the white treadmill belt.

*Clements et al., 1992*; *Rusakov and Kullmann, 1998*; *Budisantoso et al., 2013*; *Matthews et al., 2022*; *Mendonça et al., 2022*). *Table 2* details the number of glutamate traces per rat phenotype and sex included in the analyses of cue- or reward-locked maximum peak concentrations and peak numbers.

Maximum glutamate peak concentrations recorded during the cue period, and followed by turns, were significantly higher in GTs than in STs phenotype: $F(1,28.85) = 8.85$, $p = 0.006$, $\eta_p^2 = 0.23$ (*Figure 6c*). Furthermore, the presence of single peaks predominantly characterized cue-locked recordings from GTs while two or three peaks occurred relatively more frequently in STs ($X^2(2, N = $

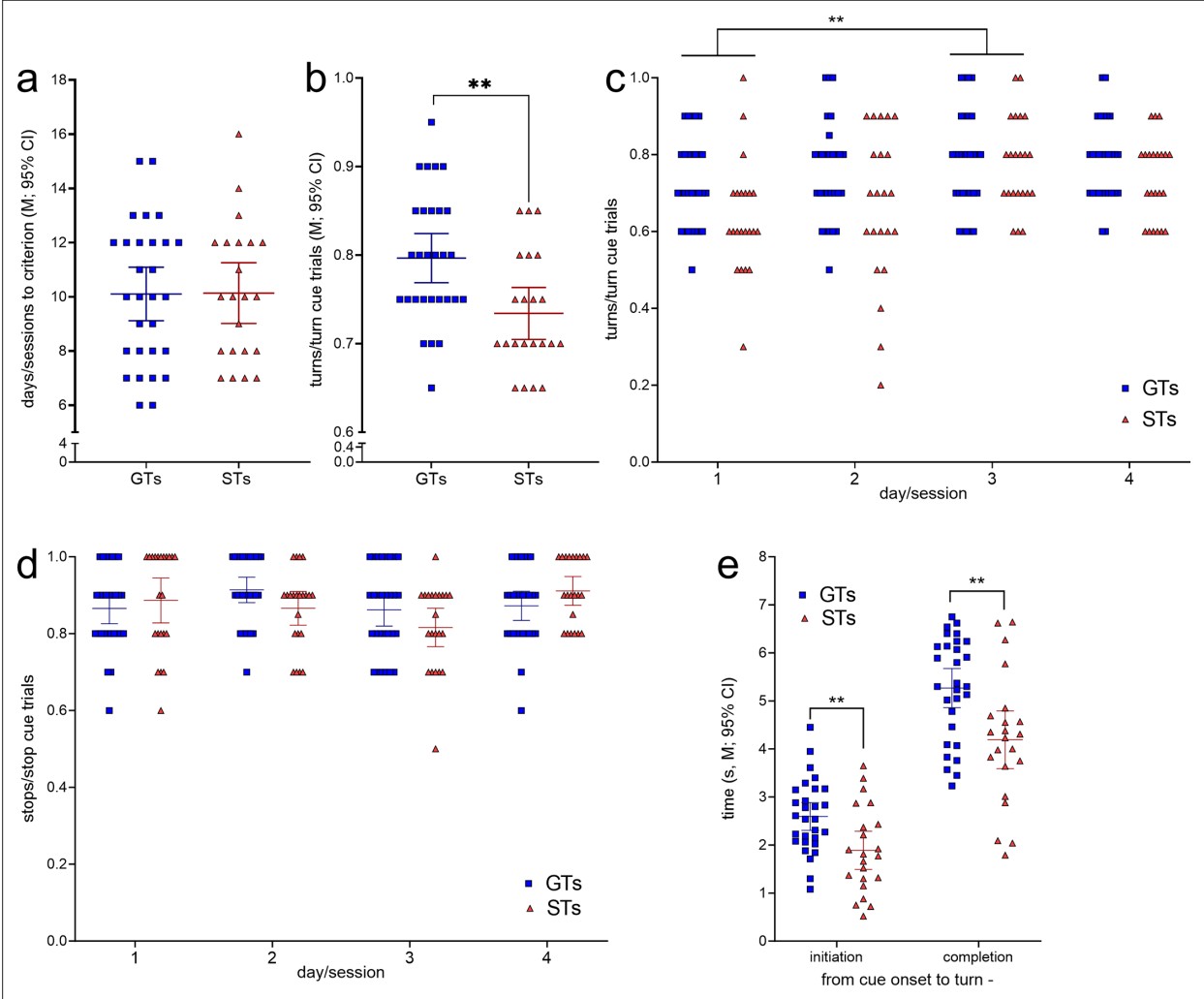

**Figure 4.** Cued-Triggered Turning Task (CTTT) acquisition and asymptotic performance by goal trackers (GTs) (*n* = 29, 13 females) and sign-trackers (STs) (*n* = 22, 12 females; see *Figure 2*). (**a**) The number of training sessions to reach CTTT criterion performance, defined as 70% correct responses to either cue for two consecutive days, differed neither between the phenotypes nor the sexes. However, baseline performance, based on data from four test sessions conducted after rats had reached criterion performance, indicated that GTs scored more cued turns than STs (**b**). Moreover, a main effect of day reflected that all rats performed more cued turns on day 3 than on day 1 (post hoc Bonferroni test (**c**); note that (**c**) and (**d**) show response ratios but ANOVAs were conducted using arcsine transformed data because ratio data violated homoscedasticity). (**d**) The relative number of cued stops did not differ between the phenotypes. There were no main effects of sex and no significant interactions between the effects of phenotype, day, and sex on cued turns and cued stops. (**e**) Relative to the time of cue onset, GTs initiated and completed cued turns later than STs. Reflecting the parallel effects of phenotype on initiation and completion times, the actual time needed to complete turns did not differ between the phenotypes (**: p < 0.01).

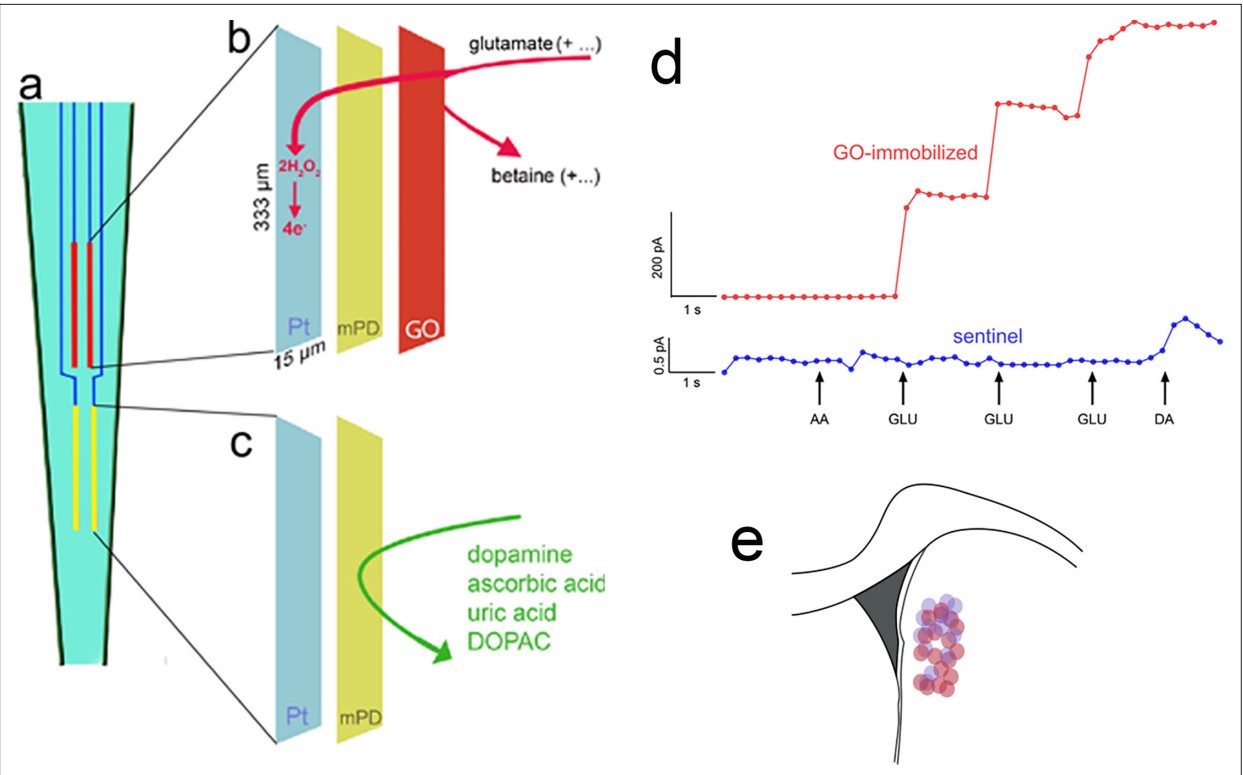

**Figure 5.** Electrochemical measurement scheme, electrode calibration, and placement. (**a**) Schematic illustration of a ceramic backbone with four platinum–iridium (Pt/Ir) recording sites, organized in two pairs, with each recording site measuring 333 µm long and 15 µm wide. The upper pair was fabricated to measure currents reflecting glutamate concentration (red, also **b**), while the lower pair served as sentinels for background current recording and subtraction (yellow, also **c**). Glutamate oxidase (GO) was immobilized onto the upper (**b**), but not lower (**c**), pair of recording sites. After GO coating, a non-conducting polymer, *m*-(1,3)-phenylenediamine was electroplated onto both sites to prevent the transfer of small electroactive organic molecules to the Pt sites. Application of a 0.7-V constant potential versus an Ag/AgCl reference electrode (not shown) served to oxidize hydrogen peroxide at individual recording sites (e.g., *Burmeister et al., 2000*; *Parikh et al., 2010*), yielding current proportional to the concentration of glutamate at the recording site. (**d**) In vitro calibration (representative example) indicated the sensitivity, linearity of the response to increasing concentrations of glutamate, and selectivity of GO-coated recording sites (red; see *Figure 2* for calibration data). The current recorded via the sentinel site did not respond to glutamate and the *meta*-(1,3)-phenylenediamine (*m*PD) barrier completely prevented ascorbic acid (AA) from generating current, Addition of dopamine (DA) increased the current by about 0.5 pA (note that the ordinate used to graph the current obtained via the GO-immobilized site (200 pA) obscures the visualization of such a small increase). For the 3/15 electrodes that exhibited a dopamine response (>0.1 pA; see *Table 1*), net currents were normalized by the dopamine response (*Burmeister and Gerhardt, 2001*). (**e**) Electrodes were implanted into the dorsomedial striatum (DMS); see Methods for stereotaxic coordinates of accepted placements; the circles in (**e**) indicate the location of microelectrode tips placed within this area (red circles: placements in rats which were recorded during Cued-Triggered Turning Task (CTTT) performance; purple circles: placements in rats in which fronto-striatal projections were silenced; note that placements along the anterior–posterior axis were flattened onto the section shown in **e**).

**Table 1.** Electrode in vitro calibration characteristics.

| | Sensitivity (slope) to glutamate (*M* ± SEM) | Selectivity for glutamate over ascorbic acid (AA; *M* ± SEM) | Limit of detection of glutamate (*M* ± SEM) | Linearity of response ($R^2$; *M* ± SEM) | Dopamine-evoked current (individual electrode values*) |
|---|---|---|---|---|---|
| Minimum criteria | >5 pA/µM | >50:1 glutamate:AA | <1.0 µM | $R^2$ > 0.95 | n/a |
| Measured (*N* = 15 electrodes; 8 in GTs, 7 in STs) | 12.0 ± 0.2 | 113.1 ± 23.5 | 0.3 ± 0 | 0.9 ± 0 | 0.26, 0.47, and 0.53 pA* |

*Recordings of glutamate reflecting currents in vivo using these three electrodes (two in goal trackers [GTs] and one in a sign-tracker [ST]) were normalized (divided) by dopamine-evoked currents.

**Table 2.** Number of glutamate traces extracted for the analysis of glutamate peaks locked to turn cues, stop cues, or reward delivery, per phenotype and sex.

Note: Trace counts from rats expressing the empty DREADD control construct are not included in this table as *Figures 6 and 7* do not show the (insignificant) results from these rats.

| Glutamate release data shown in Figure | | GTs | | STs | |
|---|---|---|---|---|---|
| | | **Females** | **Males** | **Females** | **Males** |
| 5c, 5d (turns) | Traces per rat, range | 11–13 | 12–16 | 10–14 | 12–15 |
| | Traces per rat, median | 12 | 12.5 | 12 | 13 |
| | Total number of traces | 53 | 48 | 24 | 67 |
| | Number of rats | 4 | 4 | 2 | 5 |
| 5e (misses) | Traces per rat, range | 6–9 | 7–8 | 6–9 | 6–7 |
| | Traces per rat, median | 7.5 | 8 | 7.5 | 6 |
| | Total number of traces | 29 | 32 | 12 | 34 |
| | Number of rats | 4 | 4 | 2 | 5 |
| 5f (stops) | Traces per rat, range | 12–17 | 17 | 14–18 | 11–17 |
| | Traces per rat, median | 17 | 13 | 16 | 14 |
| | Total number of traces | 57 | 58 | 29 | 76 |
| | Number of rats | 4 | 4 | 2 | 5 |
| 5g (reward delivery) | Traces per rat, range | 14–15 | 14–15 | 11–12 | 11–12 |
| | Traces per rat, median | 14.5 | 14.5 | 11 | 11 |
| | Total number of traces | 58 | 58 | 23 | 56 |
| | Number of rats | 4 | 4 | 2 | 5 |
| 7g, 7h, 7i (DREADD-expressing rats; turns after vehicle) | Traces per rat, range | 7–10 | 7–8 | 10 | 7–11 |
| | Traces per rat, median | 10 | 8 | 10 | 9 |
| | Total number of traces | 37 | 23 | 10 | 47 |
| | Number of rats | 4 | 3 | 1 | 5 |
| 7g, 7h, 7i (DREADD-expressing rats; turns after CNO) | Traces per rat, range | 2–8 | 5–7 | 8 | 7–9 |
| | Traces per rat, median | 5.5 | 6 | 8 | 8 |
| | Total number of traces | 21 | 18 | 8 | 42 |
| | Number of rats | 4 | 3 | 1 | 5 |
| 7j (DREADD-expressing rats; misses after vehicle) | Traces per rat, range | 4–7 | 4–7 | 6 | 8 |
| | Traces per rat, median | 5.5 | 4 | 6 | 4 |
| | Total number of traces | 22 | 15 | 6 | 23 |
| | Number of rats | 4 | 3 | 1 | 5 |
| 7j (DREADD-expressing rats; misses after CNO) | Traces per rat, range | 7–11 | 6 | 3 | 2–4 |
| | Traces per rat, median | 8.5 | 6 | 3 | 3 |
| | Total number of traces | 35 | 18 | 3 | 16 |
| | Number of rats | 4 | 3 | 1 | 5 |
| 7k (DREADD-expressing rats; reward after CNO) | Traces per rat, range | 10–14 | 10–16 | 12 | 11–14 |
| | Traces per rat, median | 12 | 13 | 12 | 12 |
| | Total number of traces | 36 | 52 | 12 | 62 |
| | Number of rats | 4 | 3 | 1 | 5 |

192) = 1 6.36, p = 0.0003; *Figure 6d*). Recordings during cue periods that were followed by failures to turn, or misses, indicated higher maximum peak amplitudes in STs when compared to GTs (missed turns: $F(1,193) = 55.20$, $p < 0.001$, $\eta_p^2 = 0.22$, *Figure 6e*; no phenotype effects on peak number; not shown). Likewise, relatively higher maximum peak amplitudes in STs were recorded during stop cue periods when followed by stops ($F(1,27.67) = 33.32$, $p < 0.001$, $\eta_p^2 = 0.55$, *Figure 6f*) and locked to reward delivery ($F(1,19.28) = 28.88$, $p < 0.001$, $\eta_p^2 = 0.40$, *Figure 6g*; note that errors in response to stop cues – false turns – were rare and therefore not analyzed).

## Glutamate trace characteristics predicting cued turns

The characteristics of glutamate traces (maximum peak concentration, number of peaks, and time to peak) were extracted from a total of 548 recordings of turn cue trials, 364 of which yielded a turn (GTs: 206, STs: 158) and 184 a miss (GTs: 112, STs: 72), to determine whether such characteristics, individually or in combination, disproportionally predicted more cue-triggered turns in GTs or STs. Contingency tables were used to compare phenotype-specific proportions of cued turns and misses, and to compute the probability for turns in GTs relative to STs.

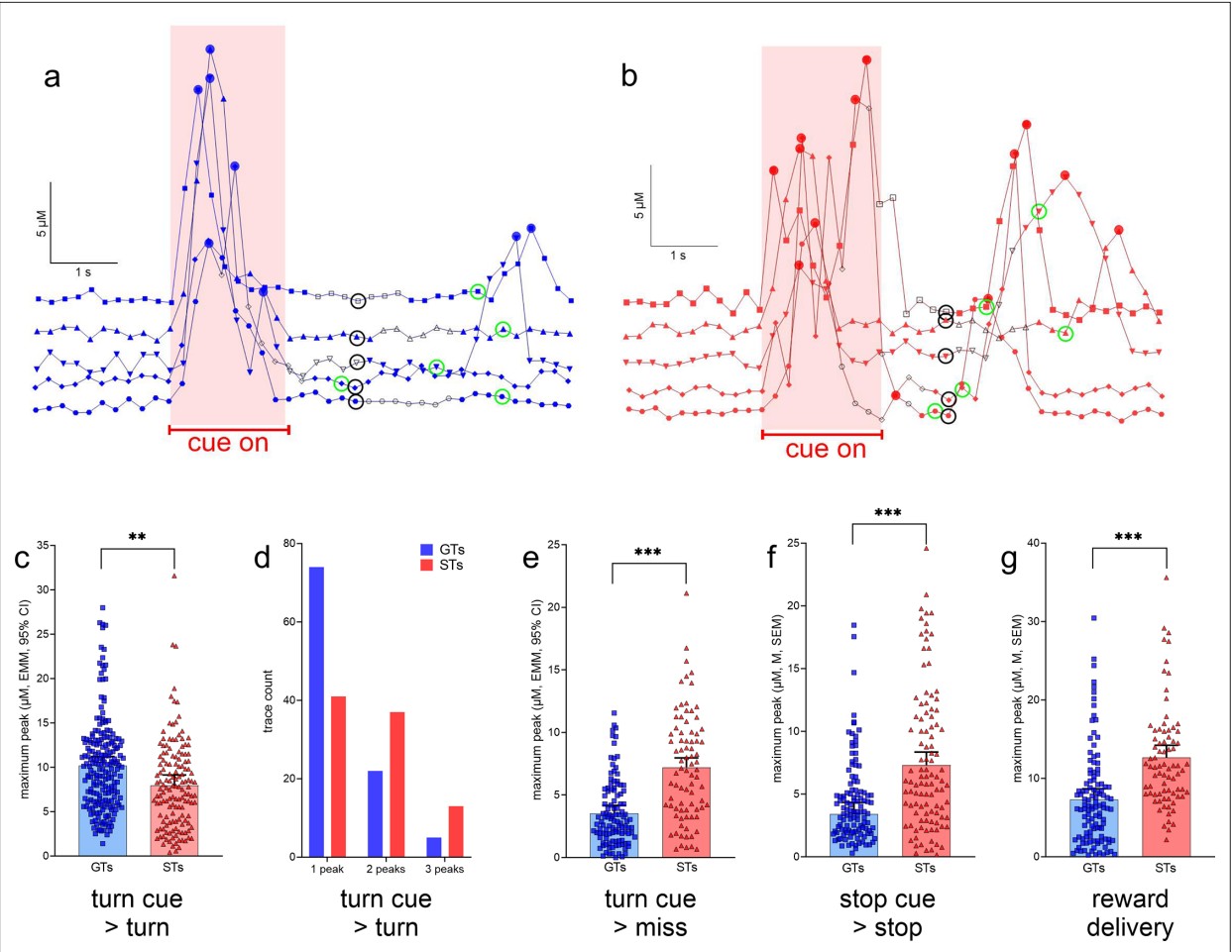

**Figure 6.** Glutamate concentrations locked to turn and stop cues and reward delivery. (**a**) and (**b**) show representative recording traces from goal trackers (GTs) and sign-trackers (STs) (five each), respectively, recorded during cued turns. Filled circles indicate peaks (defined in Methods), open black symbols were associated with the execution of the turn, symbols surrounded by a black circle with treadmill stops and the onset of a 5-s pause (see also *Figure 3*), and symbols surrounded by a green circle mark the delivery of reward following a cued turn. Currents recorded during the cue-on period (reddish background) were used to determine cue-locked peak characteristics, and those recorded during a 2-s period following reward delivery were used to compute reward-locked glutamate peaks. These current traces illustrate the predominance of single turn cue-locked peaks in GTs (**a**), contrasting with the more frequent presence of two or three turn cue-locked peaks in STs (**b**). Furthermore, reward delivery more reliably evoked glutamate peaks in STs (**b**). Linear-mixed effects model (LMM)-based analysis of glutamate traces (see *Table 2* for the number of traces per phenotype and sex included in this analyses) indicated significantly higher turn cue-locked maximum peaks in GTs when followed by actual turns (**c**). Furthermore, single glutamate peaks were observed more frequently in GTs than in STs, while two or three peaks during the cue period tended to be more frequent in STs ($X^2$ test; **d**). In contrast to glutamate currents recorded during the cue period, maximum glutamate concentrations were higher in STs during all other response categories (missed turns, **e**; cued stops, **f**; note that errors following stop cues, i.e., false turns, were extremely rare and thus peaks obtained from these trials were not analyzed), and following reward delivery during cued turn trials (**g**; **c**, **e**–**g** depict estimated marginal means (EMMs), and 95% CI; main effects of phenotype: **, ***: p < 0.01, 0.001).

Turn cue-evoked glutamate maximum peak concentrations greater than 2.8 μM were followed by disproportionally more turns in GTs than STs (2.8–8 μM: p = 0.045 to <0.0001). With increasing glutamate maximum threshold concentrations, GTs were 1.12 (>2.8 μM) to 1.40 (>8 μM) times as likely as STs to turn (middle curve, circles in *Figure 7a*; concentrations >10 μM yielded contingency table cell counts of *n* < 10 and thus were not included).

Furthermore, if one single glutamate peak occurred following the onset of turn cues and during the 2-s cue presentation period, GTs were 1.43 times as likely as STs to turn (p < 0.0001). In addition, if only one single glutamate peak followed the turn cue, increasing maximum peak concentrations predicted significantly higher relative probabilities for turns in GTs, plateauing at >4 μM where GTs

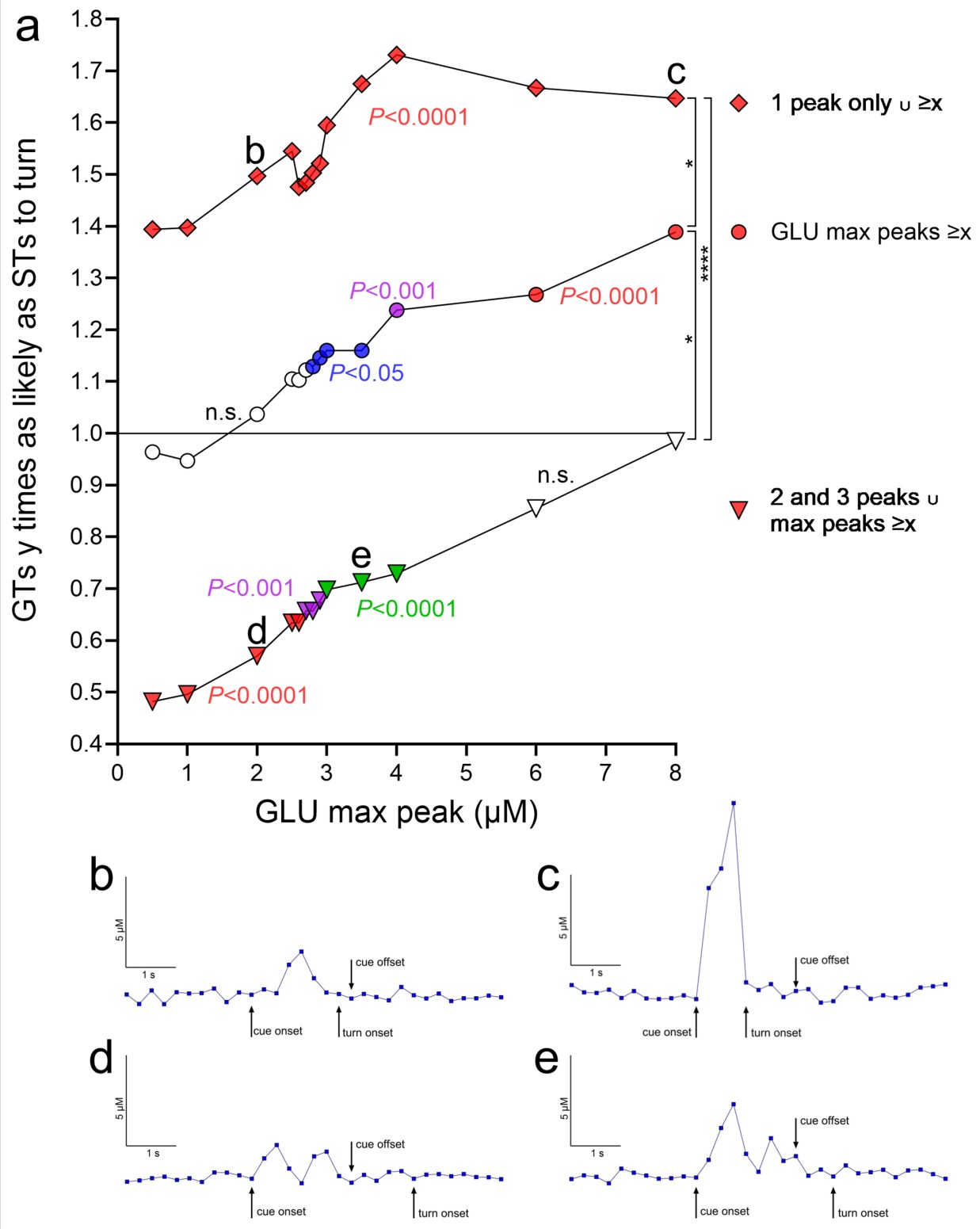

**Figure 7.** Relative probabilities for cued turns in goal trackers (GTs) given the presence of individual and combined properties of turn cue-locked glutamate peaks (based on a total of 548 traces, 364 recorded during cued turns and 184 during misses, 318 from 8 GTs and 230 from 7 sign-trackers [STs]). (**a**) The ordinate depicts relative turn probabilities derived from contingency table analyses. A value of 1 indicates that GTs were as likely as STs to turn (inserted horizontal line), while a value of 2 that GTs were twice as likely as STs to turn. For each relative turn probability value, the associated significance level, derived from contingency table analyses and reflecting the degree of dissimilarity of the proportion of turns and misses in GTs versus STs, is indicated by the symbol color (no fill, not significant, n.s.; blue, p < 0.05; green, p < 0.01; magenta, p < 0.001; red, p < 0.0001). The

*Figure 7 continued on next page*

*Figure 7 continued*

three curves in <u>a</u> depict the probability of a cued turn in GTs, relative to STs, based on: (1) turn cue-locked maximum peak glutamate concentrations (abscissa; middle curve, circles); (2) such concentrations derived from the presence of a single turn cue-locked peak (top curve, rhombi); and (3) such concentrations in conjunction with the presence of multiple (two or three) turn cue-locked peaks (bottom curve, triangles). In (**a**), the data points labeled b, c, d, and e mark data points for which representative traces are shown in (**b–e**) (all traces are from cued turns in GTs, as the plot in (**a**) indicates the probability of GTs to turn relatively to STs). For example, the data point next to the label b in (**a**) indicates that in the presence of a single glutamate peak of about 2.0 µM, GTs were 1.5 times as likely as STs to turn (significantly different from as likely to turn as STs at p < 0.0001). A trace exemplifying this data point is shown in (**b**). Concerning the significance of the three probability curves in (**a**) (1) middle curve, circles: regardless of other glutamate peak number, increasing maximum peak concentrations, beginning with 2.8 µM glutamate, yielded significantly different proportions of turns and misses in GTs and STs and rising relative probabilities for GTs to execute a turn ('GLU max peaks ≥x'; the slope of the linear regression of all three curves was significantly different from zero; see Results). For example, for maximum peak concentrations ≥4 µM, GTs were 1.24 times as likely as STs to turn. (2) Top curve, rhombi: the presence of a single turn cue-locked peak strongly increased the relative probability for GTs to turn (see Results). In conjunction with increasing amplitudes of these peaks ('1 peak only ∪ ≥x'), these probabilities did further increase, reaching, for example, 1.73 for glutamate concentrations ≥4 µM. (3) Bottom curve, triangles: the presence of two or three turn cue-evoked peaks (no more than three turn cue-locked peaks were observed) significantly lowered the relative probability for a turn in GTs below 1 or, conversely, indicated that STs were more likely to turn as GTs. However, in conjunction with rising max amplitude threshold levels ('2 and 3 peaks ∪ max peaks ≥x'), the relative probabilities of GTs to turn increased, so that at maximum peak concentrations of ≥6 µM glutamate, the proportions of turns and misses no longer differed significantly between the phenotypes. The vertical brackets on the right symbolize significant differences between the relative probabilities of the three trace characteristics ($H(3) = 33.41$, p < 0.0001; Kruskal–Wallis test; multiple comparisons: *, ****, p < 0.05, 0.0001).

were 1.73 times as likely as STs to turn (top curve, rhombi in *Figure 7a*; note that all data points of this curve reflect significant phenotype-dependent differences in the proportion of turns and misses).

In contrast to the significantly higher odds for turns in GTs following a single cue-evoked glutamate peak, following two and three glutamate peaks (no more than three turn cue-locked peaks were observed), GTs were only 0.74 times as likely as STs to turn (p < 0.0001). However, in the presence of two or three turn cue-locked glutamate peaks (peaks occurring during the cue presentation period), increasing maximum peak levels again increased the probabilities of GTs to turn, so that at maximum peak threshold levels >4 µM, GTs were as likely as STs to turn (both p > 0.85; bottom curve, triangles in *Figure 7a*). Representative traces of four selected data points are shown in *Figure 7b–e*; these traces correspond to the data points labeled b, c, d, and e in *Figure 7a*. Irrespective of whether a single or multiple glutamate peaks were evoked by the turn cue, increasing maximum glutamate concentrations increased the relative probabilities of GTs to turn. This observation was substantiated by the finding that the slopes of the linear regressions of both curves (top and bottom curves in *Figure 7a*) were different from zero (both p < 0.0001).

Together, these analyses of glutamate trace characteristics indicated that increasing maximum peak glutamate concentrations and the presence of a single, cue-evoked glutamate peak strongly increased the relative probability of GTs to execute a cued turn. Moreover, the combination of these two properties yielded significantly even higher relative turn probabilities in GTs (*Figure 7*). In contrast, the presence of multiple glutamate peaks, GTs were significantly less likely than STs to turn although, when combined with higher (<4 µM) maximum peak concentrations, turn proportions of the two phenotypes no longer differed significantly. These findings suggest fundamentally different turn cue-evoked glutamate release dynamics in GTs versus STs, perhaps involving separate afferent circuitry influencing the excitability of cortico-striatal glutamatergic terminals. Therefore, we predicted that inhibition of the cortico-striatal neurons affects turning rates and cued turn-evoked glutamate peaks primarily in GTs, consistent with prior evidence indicating their relative reliance on top–down, cortico-fugal systems to execute cued responses.

## Inhibition of fronto-striatal projection disrupts cued turning and turn cue-evoked glutamate in GTs, but not STs

As described above, in GTs, cued turning was associated with tightly orchestrated single glutamate release events that were closely locked to the turn cue, comparable with attended cue-evoked cholinergic transients in cortex (*Parikh et al., 2007*; *Gritton et al., 2016*; *Howe et al., 2017*), and therefore hypothesized to reflect cortico-striatal activation (see also Introduction). In contrast, in STs, the relatively more frequent occurrence of multiple glutamate release events and the presence of greater reward delivery-locked glutamate release may reflect a greater control of glutamate release by striatal interneuronal networks and striatal afferents originating from the

midbrain, including dopaminergic projections traditionally linked to the processing of reward. Indeed, ascending projections to the striatum may influence, directly and via striatal interneurons, the excitability of glutamatergic terminals of cortico-striatal projections (e.g., *Tritsch and Sabatini, 2012*; *Agnoli et al., 2013*; *Dautan et al., 2020*; *Moss et al., 2021*). In STs, such ascending control of glutamate release by reward circuitry may have been responsible for relatively higher reward delivery-locked glutamate levels and for the less tightly organized glutamate dynamics during cued turns. We tested the hypothesis that turn cue-locked glutamate peaks in GTs reflect cortico-striatal activation by using a dual vector approach to express an inhibitory DREADD in fronto-striatal projection neurons.

GTs (*n* = 12, 6 females) and STs (*n* = 10, 4 females; see *Figure 2*) underwent surgery to infuse an inhibitory DREADD vector into the prelimbic cortex and a Cre-expressing, retrogradely transported plasmid into the DMS (see *Figure 8* for a timeline and a schematic illustration of the dual vector strategy used to inhibit cortico-striatal projections). An additional group of GTs (*n* = 5, 2 females) and STs (*n* = 2, 2 females; *Figure 2*) received infusions of a mCherry-expressing control vector into the prelimbic cortex, in addition to the Cre-expressing plasmid into the DMS, to allow for the assessment of potential off-target effects of clozapine *N*-oxide (CNO). Following this surgery, all rats were trained to CTTT performance criterion. Thereafter, the effects of CNO or vehicle on CTTT performance were assessed (see *Figure 2* for number of rats per phenotype and sex; *Figure 8c*). The remainder of the rats (*Figure 2*) underwent a second surgery to implant glutamate recording electrodes into the DMS (*Figure 8d*), followed by the assessment of the effects of CNO on performance and glutamate transients (*Figure 8d*).

## CNO-induced inhibition of prelimbic–DMS projections disrupts cued turns in GTs

On the first of 4 days of testing of the effects of CNO or vehicle, and when the vehicle for CNO was administered, the number of turns/turn cue trials and the number of stops/stop cue trials did not differ significantly between animals without (*Figure 8c*) and with (*Figure 8d*) implanted MEAs (both *t* < 0.65, both p > 0.53, at alpha = 0.05/2). Thus, for the initial analysis of the effects of CNO on CTTT performance, data from both groups od rats were combined. The effects of CNO/vehicle in GTs and STs on cued turns and cued stops were analyzed individually, using the average score from two tests each of the effects of vehicle and CNO, and ANOVA of the effects of phenotype and treatment (vehicle/CNO; repeated measures ANOVA, with alpha = 0.05/2). The effects of phenotype and treatment interacted significantly ($F(1,20)$ = 35.99, p < 0.0001, $\eta_p^2$ = 0.64) reflecting that CNO reduced the ratio of turns/turn cue trials in GTs by nearly 50% (for multiple comparisons see *Figure 8e*). This analysis also indicated a main effect of phenotype ($F(1,20)$ = 9.45, p = 0.006) that likewise reflected the greatly suppressed turn rates in GTs treated with CNO while turn rates in STs were unaffected by CNO (*M*, SEM: GTs: 0.54 ± 0.04 turns/turn cue trials; STs: 0.71 ± 0.04). Similarly, the relatively large effect of CNO in GTs accounted for a main effect of treatment ($F(1,20)$ = 37.91, p < 0.0001; vehicle: 0.72 ± 0.03; CNO: 0.53 ± 0.03). The post hoc comparison between the turn rates of vehicle-treated GTs and vehicle-treated STs remained insignificant, reflecting that compared to the analysis of baseline performance (*Figure 4b*), GTs turned relatively less frequently when following additional surgeries to infuse the DREADD construct and implant electrodes (*Figure 8e*). Perhaps the performance of GTs was impacted relatively more severely by these surgeries, consistent with evidence indicating a relatively greater response of the innate immune system of GTs to immune challenges (*Carmon et al., 2023*).

To control for potential off-target effects of CNO, the effects of CNO (5.0 mg/kg) and vehicle on the relative number of cued turns were assessed in GTs expressing the empty control vector (*n* = 5). In these rats, CNO did not affect the relative number of cued turns ($t(3)$ = 0.78, p = 0.50; means: vehicle: 0.77 turns/turn cue trials; CNO: 0.73). Although there were no effects of CNO on cued turns in DREADD-expressing STs (above), we also tested the effects of CNO in two STs expressing the empty control vector (mean of 2 test sessions per treatment: vehicle: 0.6 and 0.6 cued turns/turn cue trials; CNO: 0.65 and 0.75). Therefore, and similar to results from previous studies (see Methods; see also *Avila et al., 2020*; *Lawson et al., 2023*), these findings did not suggest off-target effects of CNO, at the dose used, on rodent behavior.

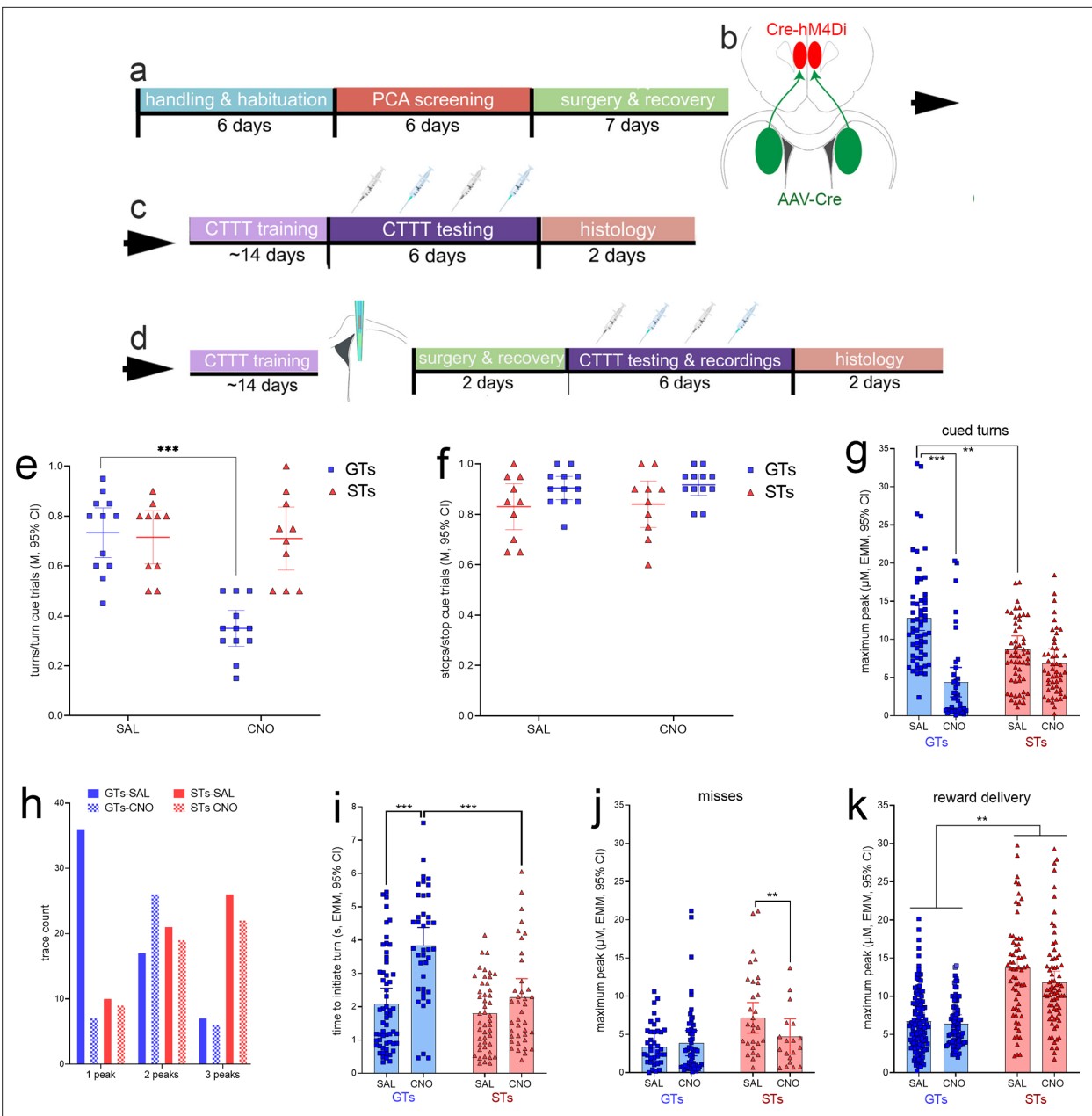

**Figure 8.** Timeline of experimental procedures and effects of clozapine *N*-oxide (CNO) on Cued-Triggered Turning Task (CTTT) performance. (**a**) Following initial handling and screening to identify the rat's phenotype (17 goal trackers [GTs], 8 females; 12 sign-trackers [STs], 6 females; *Figure 2*), rats were infused with either a Cre-dependent inhibitory DREADD or the empty control vector (both expressing mCherry; **b**) into the lower layers of the prelimbic cortex, and a retrogradely transported, Cre-expressing plasmid into the dorsomedial striatum (DMS) (expressing eGFP). These rats continued to undergo CTTT performance training, followed either by (**c**) a test of the effects of vehicle or CNO on CTTT performance (the blue syringes symbolizing CNO administration) or by (**d**) implantation of an microelectrode array for the measurement of glutamate concentrations in the DMS and following vehicle or CNO administration. Following recovery from that surgery, the effects of CNO and vehicle on performance and performance-associated glutamate concentrations were assessed. In GTs, administration of CNO significantly reduced the relative number of cued turns (**e**), but not cued stops (**f**; **e** and **f** show individual data, mean and 95% CI; post hoc multiple comparisons: ***: p < 0.001, Tukey's Honest Significant Difference test). CNO had no effects on cued turns or stops in rats expressing an empty control vector (see Results). In GTs, CNO administration attenuated cue-locked maximum peak glutamate concentrations during (residual) cued turns (**g**) and decreased the number of traces with just one glutamate peak during the cue period while increasing the frequency of two peaks during this period (**h**). Following the administration of CNO, residual turns took significantly more time to be initiated in GTs when compared with vehicle-treated GTs and CNO-treated STs (**i**). When followed by failures to turn, or misses, turn cue-locked maximum peak concentrations were not affected in GTs but reduced in STs (significant interaction between the effects of phenotype and CNO; **j**). Maximum peak glutamate concentrations locked to reward delivery again were higher in STs than in GTs, and CNO reduced these concentrations in both phenotypes (main effect of CNO, no interaction **k**); (multiple comparisons: **, ***: p < 0.01, 0.001).

## Fronto-striatal inhibition attenuates turn cue-locked glutamate peaks in GTs

Guided by the results illustrated in *Figures 6 and 7* and using identical peak definitions and extraction methods, the analyses of the effects of fronto-striatal inhibition via CNO-induced DREADD activation on glutamate transients focused on cue-locked maximum peak concentrations and the number of peaks during residual turns. LMMs were used to analyze the effects of CNO or vehicle on glutamate transients recorded from GTs and STs expressing the inhibitory hM4Di DREADD or the empty control construct (*Figure 8*; see also *Table 2*).

Administration of CNO significantly reduced turn cue-locked maximum peak glutamate concentrations in GTs, but not STs phenotype × treatment ($F(1,200.79)$ = 18.42, p < 0.001; main effect of treatment: $F(1,200.79)$ = 44.76, p < 0.001; phenotype: $F(1,9.50)$ = 0.73, p = 0.41). Multiple comparisons (shown in *Figure 8g*) confirmed that CNO significantly suppressed turn cue-locked maximum peaks in GTs ($\eta_p^2$ = 0.21), but not STs, and that following the administration of vehicle, GTs again (see *Figure 6c*) exhibited higher cue-locked maximum peak levels than STs. CNO reduced maximum peaks in GTs to statistically similar levels seen in STs irrespective of treatment (mean, SEM: GTs: 8.6 ± 0.7 µM; STs: 7.8 ± 0.7 µM; *Figure 8g*).

As seen before, cue period-based glutamate traces obtained from GTs predominately featured a single peak while over 80% of the traces recorded in STs showed two or three peaks (*Figure 8h*). Administration of CNO in GTs reduced the number of single peak and increased the number of two-peak occurrences but, in STs, did not affect the frequencies of one, two, or three peak occurrences (separate Chi-squared tests on the effects of vehicle and CNO on frequencies; GTs: $X^2(2, N = 99)$ = 17.87, p < 0.001; STs: $X^2(2, N = 107)$ = 0.03, p = 0.98; *Figure 8h*).

## Residual turns following CNO-induced inhibition in GTs

CNO robustly reduced cued turn rates in GTs (*Figure 8e*). Furthermore, we observed CNO-attenuated maximum glutamate peaks (*Figure 8g*) during (residual) cued turns, raising the question whether these residual turns differed behaviorally from those seen in vehicle-treated GTs. Therefore, we compared residual cued turns following CNO with turns executed following vehicle administration. Administration of CNO significantly increased the time GTs took to initiate turns, relative to cue onset, when compared with vehicle-treated GTs and CNO-treated STs (LMM; phenotype × treatment: $F(1,173.72)$ = 9.70, p = 0.002, $\eta_p^2$ = 0.05; see *Figure 8i* for multiple comparisons).

## Effects of CNO-mediated fronto-striatal inhibition on glutamate peaks during misses, cued stops, and reward delivery

In contrast to turn cue-locked maximum glutamate concentrations in trials followed by turns, when followed by misses, CNO had no effect (main effect of treatment: $F(1,133.99)$ = 1.62, p = 0.21; *Figure 8j*). However, a significant interaction between the effects of treatment and phenotype ($F(1,133.99)$ = 3.88, p = 0.04; $\eta_p^2$ = 0.03) reflected that CNO significantly reduced cue-locked maximum peak concentrations in STs in trials yielding misses (results of multiple comparisons are shown in *Figure 8j*). There were no effects of CNO on stop cue-locked maximum peak glutamate concentrations, irrespective of whether the stop cue was followed by a stop or a (false) turn (all p > 0.23; not shown).

As before (*Figure 6g*), peak glutamate concentrations locked to reward delivery were higher in STs than in GTs (main effect of phenotype: $F(1,10.45)$ = 22.38, p < 0.001, $\eta_p^2$ = 0.68; *Figure 8k*). CNO administration resulted in a significant decrease of reward-locked glutamate maximum peak concentrations in both phenotypes (main effect of treatment: $F(1,412.73)$ = 4.83, p = 0.03, $\eta_p^2$ = 0.01; interaction: $F(1,412.729)$ = 1.35, p = 0.25; *Figure 8k*).

As described above, administration of CNO to GTs expressing the DREADD construct attenuated maximum glutamate peaks, increased the occurrence of turn cue-locked traces with more than one peak, and slowed the initiation of the turn. In STs expressing the DREADD construct, CNO had no effects on these measures. The lack of effects of CNO on cued turns in five GTs expressing the empty control vector is described above. In addition, we extracted turn cue-locked glutamate traces (vehicle: 18 traces; CNO: 16 traces) from an empty vector-expressing GT. Administration of CNO did not reduce maximum glutamate peak concentrations (*M*, SD; vehicle: 8.51 ± 3.67 µM; CNO 8.59 ± 5.60 µM; $t(16)$ = 0.05, p = 0.96) and did not appreciably increase the proportion of traces with just one peak (vehicle: 67%; CNO: 69%). Furthermore, CNO did not slow the initiation of the actual turns

($M$, SD; vehicle: 1.80 ± 1.29 s; CNO 1.70 ± 1.19 s; $t(16)$ = 0.14, p = 0.89). The absence of effects of CNO on cued turning performance and on turn cue-locked glutamate dynamics is consistent with prior studies showing the absence of behavioral effects of 5.0 mg/kg CNO in rats not expressing the DREADD vector (*Avila et al., 2020*; *Kucinski et al., 2022*).

## DREADD expression in prelimbic cortex predicts CNO efficacy

The efficacy of the expression of eGFP in the DMS and of mCherry in the prelimbic cortex was quantified using categories that are illustrated in *Figure 9a–f*. Furthermore, we computed the proportion of neurons in the prelimbic cortex which co-expressed both fluorochrome reporters (*Figure 9h, i*). As the area of eGFP expression in neuronal soma in the DMS may only partially correspond to the synaptic space of prelimbic projections, it was not unexpected that DMS eGFP transfection efficacy scores did not correlate significantly with the efficacy of CNO to reduce cued turns (GTs: $R^2$ = 0.01, p = 0.84; STs: $R^2$ = 0.32, p = 0.09). In contrast, in GTs, but not STs, the expression efficacy of mCherry in the prelimbic cortex, that is, of the inhibitory DREADD construct, was significantly correlated with the efficacy of CNO to attenuate cued turns (GTs: $R^2$ = 0.48, p = 0.02; STs: $R^2$ = 0.06, p = 0.51; *Figure 9g*). Likewise, in GTs, but not STs, the proportion of neurons expressing both the reporter for the inhibitory DREADD and the retrogradely transported Cre-expressing construct was positively correlated with CNO-induced reduction of cued turns (GTs: $R^2$ = 0.77, p = 0.0004; STs: $R^2$ = 0.06, p = 0.51; *Figure 9j*). Thus, in GTs, the degree of CNO-induced inhibition of prelimbic cortex–DMS projections predicted the efficacy of CNO to attenuate cued turning rates.

## Discussion

The present experiments assessed the role of cortico-striatal glutamate signaling for cued movement and movement suppression. Furthermore, GTs and STs were selected for the expression of opponent cognitive-motivational biases, to test the hypothesis that cue-locked cortico-striatal glutamate signaling contributes to cued movements preferably in GTs. The results from our experiments support the following main conclusions. (1) Turn cue-evoked glutamate concentrations reached higher levels in GTs compared with STs. In contrast, glutamate concentrations during missed turns, cued stops, and locked to reward delivery, reached significantly higher levels in STs. (2) Turn cue-evoked glutamate peak concentrations, in conjunction with single spikes, nearly doubled the likelihood for a turn in GTs relative to STs. (3) In GTs but not STs, inhibition of cortico-striatal neurons reduced cued turn rates and turn cue-evoked glutamate peak concentrations, and it increased the number of glutamate peaks observed during the turn cue period. (4) In GTs, but not STs, the efficacy of CNO to attenuate cued turns was significantly correlated with transcription efficacy scores for the inhibitory DREADD, and with the relative number of prelimbic–DMS neurons expressing the inhibitory DREADD.

The impact of the behavioral phenotypes on these results can be illustrated by contrasting the actual results with hypothetical findings obtained had experiments been conducted in non-selected rats. In the absence of phenotype as a factor: (1) cue-evoked glutamate concentrations merely would have been found to be higher during turns when compared with missed turns and stops, but not reward delivery; (2) some animals would have been found to turn when peak glutamate concentrations were relatively low but multiple peaks were present, and some when the maximum concentrations of single glutamate spikes reached the same relatively low peak concentrations; consequently, the number of glutamate spikes would have been concluded to have no impact on turn probability; (3) the effects of CNO in DREADD-expressing rats would have been found to be extremely variable, reducing turning rates in some rats but not in others, but the collective results would not have offered an explanation of such a finding; (4) the correlations between, on the one hand, the DREADD transcription scores and the number of double-labeled neurons in prelimbic cortex and, on the other hand, turn rates, are insignificant (both $R^2$ < 0.17), rejecting the attribution of the behavioral and glutamatergic effects of DREADD-mediated inhibition of cortico-striatal projections. These contrasts between the actual results and these hypothetical findings illustrate an essential role of the impact of the phenotypes for the discussion of the present evidence.

Goal- and sign-tracking are behavioral indices that predict the presence of larger, opponent cognitive-motivational biases (for review see *Sarter and Phillips, 2018*). STs preferably assign motivational significance to reward cues, with the result that they perceive such cues as rewarding, worth

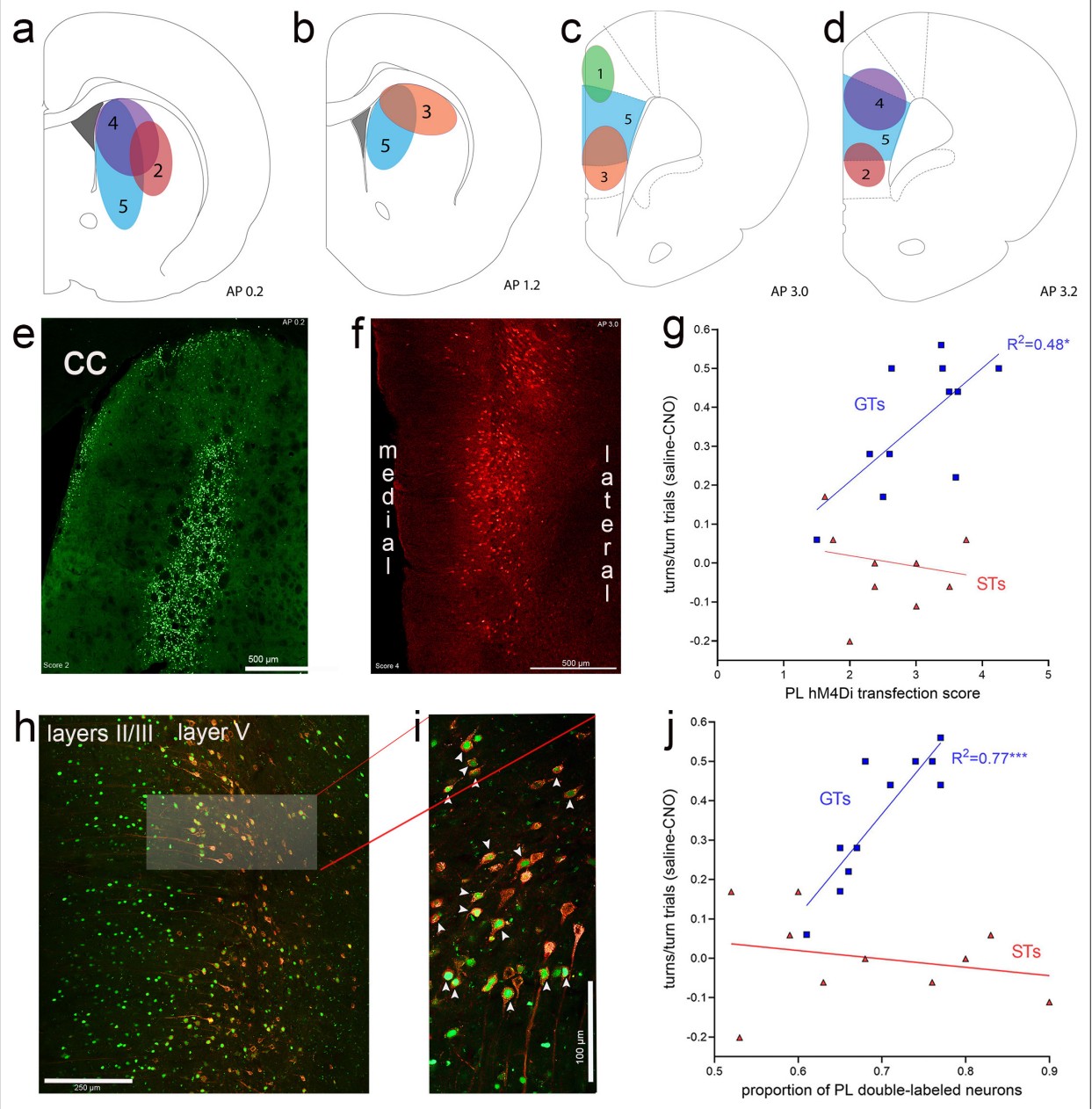

**Figure 9.** Expression of eGFP (green channel), indicating the expression of the retrograde Cre-expressing vector in the dorsomedial striatum (DMS) and, following retrograde transport, in the medial frontal cortex. The presence of the amplified mCherry fluorescent reporter signal (red channel) in cortex indicated the expression of the inhibitory hM4Di DREADD vector. Illustration of expression efficacy ratings for eGFP in the striatum (**a, b**) and of mCherry in the prelimbic cortex (**c, d**), with a top score of 5 indicating complete or near-complete expression exclusively in the DMS projection field of prelimbic cortex efferent neurons (***Mailly et al., 2013***) and prelimbic cortex, respectively. The example of eGFP expression in the DMS in (**e**) (cc, corpus callosum) received a score of 2 (illustrated in **a**) because the expression field was located in part lateral to the prelimbic cortex projection field. (**f**) depicts an example of mCherry expression in the prelimbic cortex that was assigned a score of 4 as it was restricted to the deeper layers in the prelimbic cortex and extended dorsally into cingulate cortex. In goal trackers (GTs), but not sign-trackers (STs), mCherry expression scores in cortex were significantly correlated with the efficacy of clozapine *N*-oxide (CNO) to reduce cued turns (**g**; note that the ordinate in (**g**) depicts the difference between the proportion of turns in vehicle- minus CNO-treated rats, so that higher scores indicate greater CNO effects, and therefore higher expression scores were correlated with greater CNO effects). (**h**) depicts the retrogradely transported eGFP and the expression of mCherry, including the co-expressing of both fluorochromes, primarily in layer 5 of the prelimbic cortex. The brightened rectangular region in (**h**) is enlarged in (**i**) and shows neurons co-expressing nucleolar eGFP and cytoplasmic mCherry (white arrows in **i**). In GTs, but not STs, the proportion of double-labeled neurons in prelimbic cortex was significantly correlated with the efficacy of CNO to attenuate cued turns (**j**) note that higher scores on the ordinate depicts greater CNO effects.

approaching and working for. In contrast, GTs are biased toward a (top–down) analysis of the behavioral utility of a reward or drug cue. Consistent with the interpretation of goal-tracking as indicating a bias for deploying top–down approaches, GTs were shown to be able to utilize complex occasion setters predicting the availability of cocaine (*Saunders et al., 2014*; *Pitchers et al., 2017b*). Research in humans has confirmed the presence of such broad opponent cognitive-motivational styles indexed by sign- and goal-tracking, particularly the propensity of STs to attend to cues preferably as a function of cue salience and prediction of reward (*Schad et al., 2020*; *Duckworth et al., 2022*; *Colaizzi et al., 2023*).

The present findings indicate that, in GTs, turn cue processing in the striatum of is closely controlled by cortico-striatal activity, and that they utilize turn cues more reliably than STs. These findings complement prior evidence from experiments assessing the selection of cues for extended processing and behavioral control. GTs out-performed STs in tasks taxing the attentional detection of cues, in part by deploying goal-directed attentional mechanisms to detect cues and maintain stable task performance (e.g., *Paolone et al., 2013*; *Kucinski et al., 2018*; *Phillips and Sarter, 2020*; *Kucinski et al., 2022*). The present results are consistent with a relatively less efficacious cue detection process in STs, mediated via their relatively deficient cortical cholinergic mechanisms (*Paolone et al., 2013*; *Koshy Cherian et al., 2017*; *Pitchers et al., 2017a*; *Kucinski et al., 2022*; *Carmon et al., 2023*) and, consequently, and via cortico-striatal projections, a relatively less effective utilization of cues to control movements.

In GTs, inhibition of prelimbic–DMS projections attenuated cued turns and associated glutamate concentrations. In contrast, suppression of movement, as seen during cued stops, and associated glutamate release dynamics, remained unaffected by CNO and, therefore, may not depend on cortico-striatal processing of movement cues (see also *Vandaele et al., 2019*; *Cruz et al., 2022*). In GTs, turn cue-evoked glutamate concentrations as well as the presence of single peaks may be speculated to depend, or be secondary to, the cue detection-mediating generation of single cholinergic transients in prelimbic cortex and the resulting generation of high-frequency oscillations (*Gritton et al., 2016*; *Howe et al., 2017*; see also Introduction). In the DMS, single high-amplitude glutamate transients from cortical afferents may recruit multiple populations of interneurons and striatal output neurons, thereby forming, for a precise period of time, fronto-striatal cell assemblies to force cue-associated action selection (see also *Hart et al., 2018a*; *Hart et al., 2018b*; *Oberto et al., 2022*).

Cortico-striatal glutamatergic terminals are also recruited, directly and indirectly, via striatal interneurons and striatal dopaminergic afferents, thereby supporting bidirectional glutamatergic–dopaminergic interactions and allowing reward expectation and action outcome to influence the glutamatergic representation of movement cues (*Bamford et al., 2004*; *Agnoli et al., 2013*; *Kosillo et al., 2016*; *Cai and Ford, 2018*; *Moss et al., 2021*; *Chantranupong et al., 2023*; *Choi et al., 2023*; *Krok et al., 2023*; *Holly et al., 2024*). Such striatal control of cortico-striatal terminals, primarily via cholinergic and dopaminergic heteroreceptors (for review see *Lovinger et al., 2022*), may preferentially yield suppression of action in STs. This view is consistent with several current findings and considerations. (1) Peak glutamate concentrations in STs were relatively higher during stops, irrespective of the validity of stops (false stops or stop cue-evoked stops). (2) The presence of multiple cue-locked glutamate peaks in STs (a second or third peak occurred on average 1.58 ± 0.39 s (*M*, SD) and 1.83 ± 0.25 s following the onset of the turn cue), may reflect relatively slow, reverberating interactions between striatal interneurons, dopamine signaling, and cortico-striatal glutamate release (e.g., *Dorst et al., 2020*; *Frost Nylén et al., 2021*). In GTs, the relatively rare presence of multiple peaks during turn cue periods may have interfered with cortico-striatal cue import and therefore reduced the efficacy of cued turning. (3) Inhibition of the prelimbic–DMS pathway impacted neither turn cue-locked glutamate release in STs nor cued stops or stop cue-locked glutamate release in either phenotype. (4) Reward delivery-locked peak glutamate concentrations were higher in STs than GTs. Thus, in STs, glutamatergic transients may be speculated to primarily signal the expectation of reward, and also the heightened perceptual value of a reward predicting cue (see also *Pitchers et al., 2017a*; *Saunders et al., 2018*; *Iglesias et al., 2023*; *Bernklau et al., 2024*; *Cai et al., 2024*).

The present results contribute to bio-behavioral conceptualizations of the impact of individual differences in the vulnerability for compulsive drug taking (e.g., *Volkow et al., 2006*; *Robbins et al., 2012*; *Marhe et al., 2013*; *Kilts et al., 2014*; *George and Koob, 2017*; *Pitchers et al., 2018*). In contrast, the clinical significance of cognitive-motivational endophenotypic traits such as goal- and

sign-tracking for the severeity of neurological disorders has yet to be explored (but see *Deik et al., 2012*; *Kucinski et al., 2018*). Consistent with the view that striatal pathology contributes little to the explanation of individual, clinical severity of PD (*Johansson et al., 2024*), our findings predict a greater vulnerability of PD patients who preferentially utilize cortico-striatal mechanisms to process movement cues, that is, GTs, for cholinergic loss-associated impairments in cued movement control and for the propensity for falls (*Sarter et al., 2014*; *Sarter et al., 2021*; *Albin et al., 2022*). In contrast, STs may generally be less efficient in executing cued movements, particularly in situations featuring complex and competing movement cues (e.g., *Beaulne-Séguin and Nantel, 2016*). The investigation of the impact of endophenotypic indices that predict individual differences in fundamental brain functions may be useful to reveal potently predictive neuronal and cognitive markers of psychiatric and neurological disease vulnerability.

## Materials and methods

### Subjects

378 Sprague Dawley rats (215 females; 250–500 g; obtained from Inotiv, West Lafayette, IN, and Taconic, Rensselaer, NY) were individually housed on a 12-hr light/dark cycle (lights on at 7:00 AM) at ~21°C with ad libitum access to food (Laboratory Rodent Diet 5001, LabDiet) and water. The experiments detailed below used four separate cohorts of rats (each consisted of 52 rats, 26 females and composed of rats obtained from both Taconic and Inotiv). The four cohorts were obtained and formed across 19 months, reflecting the experimental demands of amperometric recordings. All experimental procedures were approved by the University Committee on the Use and Care of Animals at the University of Michigan (protocol # PRO00010749 and PRO00011037) and carried out in laboratories accredited by the Association for Assessment and Accreditation of Laboratory Animal Care AAALAC; Unit # 000285; PHS assurance # D16-00072 (A3114-01). Animals acclimated to housing quarters for 2 days before the onset of experimental procedures. *Figure 2* depicts the use of a total of 378 rats across the four main experimental stages described in this report.

### Behavioral phenotyping

#### Apparatus, PCA testing, and measures

PCA-based behavioral phenotyping was carried out using conditioning chambers and methods detailed in previous reports (e.g., *Pitchers et al., 2017a*; *Pitchers et al., 2017b*; *Campus et al., 2019*; *Kucinski et al., 2022*; *Carmon et al., 2023*). Briefly, the chambers were equipped with a food port located 2.5 cm above the floor in the center of the intelligence panel, a red house light located on the opposite wall (on throughout training sessions), and a retractable lever (Med Associates) located 2.5 cm to the left or right of the food receptacle and 6 cm above the floor. The retractable lever was illuminated when extended with a white light-emitting diode placed inside the lever house. The pellet dispenser (Med Associates) delivered one 45 mg banana-flavored sucrose pellet (Bio-Serv) into the food magazine port at a time. A head entry was recorded each time a rat broke the infrared photobeam located inside the food magazine port. Conditioning chambers were situated in sound-reducing enclosures. Data collection was controlled by Med-PC IV Behavioral Control Software Suite.

Trials began by illuminating and extending the lever (conditioned stimulus; CS) for 8 s. Following lever retraction, a banana pellet was delivered into the magazine port (unconditioned stimulus) and a variable intertrial interval (ITI; 90 ± 60 s) started. A training session consisted of 25 trials and lasted 35–40 min. The following measures were extracted: (1) number of lever deflections (contacts); (2) latency to first lever deflection; (3) number of head entries into the food magazine port (referred to as food cup entries) during the presentation of the CS; (4) latency to the first magazine port entry after the CS presentation; (5) number of magazine port entries during the ITI.

#### Phenotype classification

The PCA score was computed to index the animals' propensity to approach the lever CS versus the food magazine port during the CS period. This score combined the averages of three measures obtained during the fourth and fifth day of testing (e.g., *Robinson and Flagel, 2009*; *Meyer et al., 2012*; *Yager et al., 2015*; *Pitchers et al., 2017a*; *Pitchers et al., 2017b*): (1) the probability of contacting either the lever CS or food magazine port during the CS period (P(lever) − P(food port));

(2) the response bias for contacting the lever CS or the food magazine port during the CS period: (# lever CS contacts − # food magazine port contacts)/(# lever CS contacts + # food magazine port contacts); and (3) the latency to contact the lever CS or the food magazine port during the CS period: (food magazine port contact latency − lever CS contact latency)/8. PCA scores ranged from −1.0 to +1.0, where +1.0 indicates an animal made a sign-tracking lever response on every trial and −1.0 a goal-tracking food port response on every trial. Rats with an averaged PCA index score ranging from −1.0 to −0.5 were classified as GTs, and rats with a PCA index score between +0.5 and +1.0 as STs.

## Cued-Triggered Turning Task

### Apparatus

The CTTT apparatus, behavioral methods, and the CTTT performance measures were previously described in *Avila et al., 2020*. Briefly, a treadmill, 32.51 cm wide by 153.67 cm long, was constructed by modifying a 2200 Series Flat Belt End Drive Dorner conveyor (Dorner, WI). The conveyor belt surface material was polypropylene, friction-resistant, and easily cleaned with ethanol or soap. A Dorner Variable Speed Controller (Dorner, WI), with speeds ranging from 0 to 32 cm/s or 19.2 m/min, was used to operate the conveyor. Furthermore, the conveyor was enclosed by a Faraday cage (145.10 cm long, 32.51 cm wide, and 38.1 cm tall), grounded to block out static electric fields. Copper reward ports were installed on either end of the treadmill to allow delivery of 45 mg banana pellets. A 28 V DC, 100 mA stimulus light was approximately 2.54 cm in diameter with a flat lens and mounted on both sides lengthwise on the Faraday cage (MedAssociates, Inc, St. Albans, VT). A Mallory-SonAlert audible device, mounted in the center of the Faraday cage, was used to generate the auditory cue. This device emitted a continuous tone at 68 dBA sound pressure level, which is within the acceptable range for chronic presentations, to neither elicit a fearful response nor induce hearing loss (*Turner et al., 2005*; *Castelhano-Carlos and Baumans, 2009*). Sessions were videotaped with four web cameras (Logitech C920x HD Pro Webcam, Full HD 1080p/30fps) and relayed to an Intel Xeon workstation (Dell, Round Rock, TX) via USB cords and processed with OBS Studio (Open Broadcaster Software, free and open-source software).

### Training regimen

Rats were trained to walk on a treadmill until the onset of one of two cues (tone or light, presented for 2 s), indicating that following a 5-s treadmill stop, 1 s after cue onset, the treadmill restarted in the reverse or same direction (*Avila et al., 2020*). Rats were trained using either the tone as the turn cue and the light as the stop cue, or vice versa. Cued turns and cued stops were rewarded by manually delivering a banana-flavored sucrose pellet (45 mg; Bio-Serv) into the reward port. Cued turns and cued stops were rewarded at the port located in front of the rat following completion of the turn or the stop, respectively. Rats first underwent a treadmill acclimation regimen using a procedure adapted from *Arnold and Salvatore, 2014*. For 1 week, rats were placed into the Faraday cage for 3 min with the belt paused and then acclimated to walk on the treadmill at speeds up to 9.6 cm/s or approximately 6 m/min (treadmill speed was gradually icreased during this period as detailed in *Avila et al., 2020*). Between sessions and subjects, the Plexiglass walls of the Faraday cage and the treadmill belt were wiped down with 70% ethanol and soap.

   Contrary to the acclimation phase, during which experimenters manually controlled the treadmill, the subsequent training phase was controlled entirely by custom scripts using Med-PC software and interface (MedAssociates). Rats walked at a speed of 9.6 cm/s and cues were presented for 2 s. The modality of the turn and stop cues was counterbalanced across animals. Maximally two successive presentations of a cue of the same modality were allowed. The ITI was 60 ± 45 s. The total number of trials per daily session was 18, with a maximum duration of a test session of 20 min. Upon reaching the performance criterion, after 2–3 weeks of training, defined as 70% correct responses to either cue for two consecutive days, rats were tested and videotaped for four additional days. Data from these four sessions were used to determine potential phenotype and sex-based differences in baseline performance. Thereafter, rats underwent surgery for chronic MEA implantation.

### CTTT performance measures

The number of cued turns, missed turns, cued stops, and false turns were extracted from offline scoring of session videos. For a cued turn to be scored, the animal must have initiated a turn, defined

by a rotation of the longitudinal orientation of the body of at least 90°, before the treadmill restart, that is, within 8 s of the cue onset. A stop was defined as a cessation of forward movement. For stops occurring before the treadmill stopped, rats would typically stop while positioned at the backend of the treadmill so that following a stop, the treadmill would transport them to the front of the treadmill (within the remaining 1–3 s until the treadmill stopped), without the rat contacting the front end of the test chamber. Following the treadmill stop and during the 5 s pause, a false turn was scored if the animal (falsely) turned. The time of initiation and completion of cued turns, relative to the onset of the cue, were extracted from session videos. For the analysis of baseline CTTT performance, and because rats generated a variable number of cued turns during the four sessions used for this analysis, individual turn onset and completion times were averaged across the test sessions. Additional performance measures were extracted from training and testing sessions to explore relationships between cued turning and stopping performance, phenotype and, subsequently, glutamatergic transients (detailed in Results).

## Amperometric recordings of analysis of glutamate currents

### Electrode preparation and calibration

Extracellular glutamate concentrations were measured on electrode surfaces featuring immobilized glutamate oxidase and by performing fixed-potential amperometry to determine hydrogen peroxide concentrations resulting from catalyzed oxidation of glutamate. Based on calibration curves generated in vitro, the resulting currents were expressed as micromolar glutamate concentrations at the recording site (*Burmeister and Gerhardt, 2001*; *Rutherford et al., 2007*; *Hascup et al., 2008*; *Parikh et al., 2008*; *Parikh et al., 2010*; *Parikh et al., 2014*; *Clay and Monbouquette, 2018*; *Bermingham et al., 2022*). The configuration consisted of a ceramic backbone probe (Quanteon LLC, Nicholasville, KY, USA) housing four recording sites, each 15 × 333 µm, crafted from platinum–iridium. These recording sites were grouped into two pairs, separated by 30 µm between each member of a pair and a 100-µm vertical spacing between the pairs, so that the two pairs of electrodes were linearly arranged along the shank of the probe. MEAs were modified for in vivo recordings in behaving rats by soldering four enamel-coated magnet wires (30 ga) to the terminals on the electrode panel and the other end to gold-pin connectors. Reference electrodes were constructed by soldering Ag/AgCl reference electrodes prepared from 0.008″ silver wire (A-M Systems, Carlsberg, WA) to gold-pin connectors. The pins were then inserted into a 9-pin ABS plug (GS09PLG-220, Grinder Scientific) and adhered to the microelectrode with epoxy. Custom 9-pin ABS plugs were also printed with a STUDIO G2 3D printer (BigRep, Berlin, Germany). The top pair of electrodes, termed active sites, were coated with recombinant L-glutamate oxidase (US Biological Life Sciences) solution (1 U in 1 µl DiH$_2$O), cross-linked with a bovine serum albumin and glutaraldehyde mixture, by manually applying microdroplets using a 1-µl Hamilton syringe. The lower pair of electrodes, termed sentinels, were coated with only the bovine serum albumin and glutaraldehyde mixture, to record currents unrelated to glutamate concentrations. After a minimum 24-hr incubation period to ensure optimal adherence of the enzyme layer, an exclusion layer composed of *meta*-(1,3)-phenylenediamine (*Mitchell, 2004*) was electroplated onto each recording site's surface by 5 min application of 0.85 V versus a silver/silver chloride (Ag/AgCl) reference electrode (Bioanalytical Systems). This exclusion layer prevents sensing electroactive interferents such as ascorbic acid (AA) and catecholamines (e.g., *Mitchell, 2004*). After another 24-hr incubation period for the electrode sites to dry, recording sites were calibrated to determine the sensitivity for glutamate, selectivity for glutamate versus interferents, stability, and limit of detection of glutamate. Calibrations were conducted using a FAST-16 electrochemical system (Quanteon). A constant voltage of 0.7 V was applied versus an Ag/AgCl reference electrode, and the system was placed in a heated 40 ml bath of 0.05 M PBS. Following a 30-min baseline recording, aliquots of stock solutions of AA (20 mM), glutamate (20 mM), and dopamine (2 mM) were added to the calibration beaker such that the final concentrations were 250 µM AA, 20, 40, and 60 µM glutamate, and 2 µM dopamine (see also *Wassum et al., 2012*; *Malvaez et al., 2015*). Changes in amperometric current at individual electrode sites were measured after each solution to calculate the slope (sensitivity), the limit of detection, selectivity for AA and dopamine, and linearity ($R^2$). Glutamate sensors were required, at a minimum, to have a sensitivity of >5 pA/µM glutamate, a limit of detection <1.0 µM glutamate, glutamate:AA selectivity ratio >50:1, minimal changes in current across all channels following dopamine addition (<3 pA), and a linear response to increasing glutamate concentrations (20–80 µM glutamate) of $R >$

0.95. The characteristics of the electrodes used in the present experiments exceeded these minimum requirements (*Table 1*).

## Chronic implantation of MEAs

Upon having reached stable, criterion-level CTTT performance (≥70% cued turns and stops for two consecutive days), MEAs were chronically implanted into the DMS. Rats were anesthetized using isoflurane gas (5% induction and 1–3% maintenance) and mounted on a stereotaxic frame on a heating pad to maintain a 37°C body temperature. Ophthalmic ointment lubricated eyes. A craniotomy and durotomy using a bent 27-gauge needle for dura removal were performed above the right DMS (AP: +0.50 mm; ML: −2.20 mm from bregma). Three stainless steel screws were threaded into the cranium. The MEA was then lowered 4.5 mm dorsoventrally from the dura into the striatum. At the same time, an Ag/AgCl reference electrode was implanted at a remote site in the contralateral hemisphere. The microelectrode assembly was anchored with methyl methacrylate dental cement, and exposed regions of the skull were filled with a translucent, medium-viscosity silicone adhesive to minimize leakage of the dental cement onto the brain. Animals rested over a 48-hr recovery period before moving to the next phase of the experiment. Amperometric recordings were collected by connecting the head stages to a FAST-16 potentiostat/data electrochemical system (Quanteon) via a shielded cable and low-impedance commutator. The hydrogen peroxide by-product of glutamate oxidase-mediated catalyzation of the oxidative deamination of glutamate was electrochemically oxidized by applying 0.7 V versus the Ag/AgCl reference electrode and digitized at a sampling rate of 5 Hz. Behavioral sessions began following a 50-min baseline recording period. Consistent with prior evidence showing that implanted MEAs maintain stable sensitivity and reliability for at least 7 days after implantation, electrochemical recordings were completed within 7 days of MEA implantation (*Rutherford et al., 2007*).

## Amperometry data processing and analysis of glutamate peaks

### Baseline correction and normalization

Electrochemical recording data were processed using a custom MATLAB (MathWorks) script. The background current recorded on sentinel sites was subtracted from the current recorded on the active sites, normalized by the response to dopamine (*Burmeister and Gerhardt, 2001*), and then converted to glutamate concentration based on calibration curves. Normalization of net currents by dopamine currents corrected for variations in the efficacy of the *meta*-(1,3)-phenylenediamine barrier (*Burmeister and Gerhardt, 2001*). Such normalization was only computed for recordings with electrodes which responded to dopamine (3 out of a total of 15 electrodes; see *Table 1*). Trial-associated glutamate concentrations were determined for three periods:

1. Baseline period: The concentration of glutamate currents recorded over 2.5 s prior to cue onset served as a baseline for determining the number and levels of task event-locked peaks. Therefore, glutamate concentrations shown in graphs depict sentinel-corrected, normalized (where applicable), and baseline-corrected levels. Across trials and entire sessions, only minimal drift of baseline glutamate levels was observed; however, baseline-based corrections further reduced the impact of potential minute-based variations in glutamate concentrations.
2. Cue presentation period: Glutamate concentrations recorded throughout the 2-s presentation of cues were used to determine cue-locked peak characteristics.
3. Reward delivery period: Currents recorded during a 2-s period following reward delivery were used to determine reward delivery-locked peaks.

The analysis of event-locked glutamate concentrations focused on determining maximum peak concentrations and the number of glutamate peaks. Prior studies on the effects of depolarization of synaptic terminals, blocking such depolarization, or of pharmacological manipulations of the excitability of terminals, confirmed that peak concentrations of extracellular glutamate indicate the degree and extent of terminal depolarization (*Hascup et al., 2008*; *Parikh et al., 2008*; *Parikh et al., 2010*; *Mattinson et al., 2011*; *Quintero et al., 2011*; *Parikh et al., 2014*).

### Peak identification criteria

The PeakDet function in MATLAB (MathWorks) was utilized to identify peaks, requiring setting threshold and minimum differences.

1. Threshold for peak identification: To be identified as a peak, glutamate concentrations needed to be 3 SDs above the average baseline current (recorded over 2.5 s before cue presentation).
2. Minimum differences: For a value to be identified as a peak, the preceding and subsequent values needed to be at least 1 SD (derived from baseline data) below the peak value.
3. Adjacent points: If multiple adjacent points were above 3 SDs of the baseline but within 1 SD from each other, the highest of these points was counted as a peak. In the absence of this additional criterion for determining peaks, none of such multiple, adjacent peaks would be counted, as none would have been bordered on both sides by points 1 SD below peak value (second criterion). Furthermore, applying this third criterion to such situations limited an escalation of peak counts during relatively sustained elevations of currents. Only 41 out of 752 traces (5.5%) required the application of this third criterion.

These criteria for identifying peaks were adopted from peak analyses employed in neurophysiological studies (e.g., *Ghosh et al., 2022*) and ensured a robust separation of peaks from potentially noisy recordings of relatively small currents and the false identification of peaks during fluctuations of relatively higher currents.

The maximum peak concentration (μM) and number of peaks were extracted from each trace used for the final analyses. Peak amplitude was defined as the highest glutamate concentration reached within 2 s from the onset of a cue or from reward delivery. During missed turns, peaks were not identified in a small fraction of trials (<2% in both phenotypes), yielding missed data for this measure. The number of peaks was counted across a 2-s period from cue onset or reward delivery.

## Trace inclusion and exclusion criteria

For each response category (cued turns, misses, cued stops, false turns), glutamate traces were included into the final analyses if: (1) if the electrode met the in vitro calibration criteria; (2) traces were devoid of major electrostatic interferences (such as resulting from the headstage contacting the reward port); (3) task compliance was apparent (animals walking continuously counter the direction of the treadmill and promptly reversing direction upon cue or treadmill onset); (4) the accuracy of the electrode placement was confirmed, determined following the completion of experiments. The data from one rat that developed seizures after the second day of recordings were completely excluded from the final analyses. As a result, 1–27 (range) traces per rat and response category were included in the final analyses. *Table 2* details the range and median of the total number of traces, per rat phenotype and sex, which were included in the final analyses of glutamate peaks.

## Verification of MEA placements in the DMS

MEAs were chronically implanted into the DMS to record extracellular glutamate levels while performing the CTTT. Following the completion of experiments, administration of a lethal dose of sodium pentobarbital (270 mg/kg, i.p.) was followed by transcardial perfusion of saline, followed by 4% paraformaldehyde in 0.15 m sodium-phosphate solution, pH 7.4. Extracted brains were postfixed in 4% paraformaldehyde for 24 hr, then submerged in 30% sucrose solution until they sank. Using a freezing microtome (CM 2000R; Leica), 35 μm thick brain slices were sectioned and stored in cryoprotectant until further histologic processing. Sections were mounted and processed with a Cresyl Violet Nissl stain to verify placements. A Leica DM400B digital microscope was used to photomicrograph the sections at ×1.25 and ×5. For electrochemical recordings to be included in the final analyses, microelectrodes needed to be placed within the following stereotaxic space: AP: −0.3 to 0.6 mm, ML: 2 to 2.5 mm, and DV: −4.2 to −5 mm.

## Intracranial infusions of a Cre-dependent DREADD and a retrograde Cre-vector

We utilized a pathway-specific dual vector chemogenetic strategy (*Hart et al., 2018b*) to selectively inhibit the activity of fronto-cortical projections to the DMS. Intracranial infusions of a Cre-dependent Designer Receptor Exclusively Activated Only by Designer Drug (DREADD) and a retrogradely transported Cre-expressing plasmid were carried out following PCA screening and prior to CTTT acquisition training. In a subset of rats, a second surgery, carried out following the acquisition of the CTTT, was conducted to implant MEAs into the DMS. The viral vectors containing the Cre-dependent plasmid pAAV-hSyn-DIO-hM4D(Gi)-mCherry (AddGene #44362-AAV8; titer of 2.1 × 1013 GC/ml),

Cre-dependent control plasmid pAAV-hSyn-DIO-mCherry (AddGene #50459-AAV8; titer of 2.2 × 1013 GC/ml), or the retrogradely transported Cre-expressing plasmid pENN-rAAV-hSyn-HI-eGFP-Cre-WPRE-SV40 (AddGene #105540-AAVrg; titer of 1.9 × 1013 GC/ml) were infused into the lower layers of the prelimbic cortex (for the anatomical organization of fronto-striatal projections see, e.g., *Mailly et al., 2013*) and the DMS, respectively.

Craniotomies and durotomies were carefully performed above four sites of the DMS using a bent 27-gauge needle for dura removal to minimize the impact on the subsequent implantation of recording electrodes and electrochemical recordings. One µl of pENN-rAAV-hSyn-HI-eGFP-Cre-WPRE-SV40 vector was infused (bolus) into the DMS at two sites per hemisphere (AP: +0.2/1.2; ML: ±2.5/2.2; DV: −4.5 mm from dura) to retrogradely transfect afferent projections. In addition, 1 µl of AAV-hSyn-DIO-hM4D(Gi)-mCherry or AAV-hSyn-DIO-mCherry (control vector) was infused (bolus) into the prelimbic cortex (AP: +3.2; ML: ±0.7; DV: −3.5 mm from dura) to allow for the selective inhibition of cortico-striatal projections. The injector was left in place for 8 min to minimize diffusion into the injector tract. Rodents recovered for 1 week before treadmill and CTTT training.

## Clozapine *N*-oxide

CNO was obtained from Tocris Bioscience (Bristol, United Kingdom) and dissolved 10 mg/ml in 6% DMSO in 0.9% NaCl solution. CNO (5.0 mg/ml/kg; i.p.) or vehicle was administered i.p. 50 min before the onset of CTTT testing. In rats also equipped for electrochemical recording of DMS glutamate levels, CNO was given just prior to the onset of recording the pre-task baseline (above). Rats were given vehicle and CNO on alternate days. Regarding potential off-target effects of clozapine (*Gomez et al., 2017*), we and others have previously consistently failed to detect effects of this dose of CNO in rats expressing the control vector, including in rats performing the CTTT, complex movement control tasks, or an operant sustained attention task (*Jendryka et al., 2019*; *Avila et al., 2020*; *Kucinski et al., 2022*). Given effective dose ranges of clozapine in rodents performing complex behavioral tasks (*Martinez and Sarter, 2008*), and given the proposed conversion rate of CNO to clozapine, an approximately 50- to 100-fold higher dose of clozapine would be required to produce significant effects on rodent behavior (see also *Mahler and Aston-Jones, 2018*; *Lawson et al., 2023*).

## Visualization and quantification of GFP/mCherry-expressing neurons

We amplified the mCherry fluorescent reporter signal of the inhibitory hM4Di DREADD vector to enhance the evaluation of the transfection efficacy and distribution of neurons co-expressing mCherry and eGFP. The eGFP fluorescent label did not necessitate signal enhancement. Sections underwent six washes for 5 min each in 0.1 m PBS, pH 7.3, and then were immersed in 0.1% Triton X-100 diluted in PBS for 15 min. After three 5 min PBS rinses, sections incubated for 60 min at room temperature in the blocking solution, 1% normal donkey serum and 1% Triton X-100 made in PBS. Sections were incubated overnight in the primary antibody (rabbit anti-mCherry, ab167453, Abcam; 1:500; diluted in blocking solution to prevent non-specific binding). The next day, and following three 5 min PBS rinses, sections were incubated for 90 min at room temperature in the secondary antibody (donkey anti-rabbit conjugated to Alexa 594, PIA32754, Invitrogen, 1:500). Following three 5 min rinses with PBS and sections were mounted, air dried, and cover-slipped with Vectashield Antifade Mounting Medium (H-1000; Vector Laboratories).

A Zeiss LM 700 confocal microscope, equipped for sequential multi-track acquisition with 488- and 561-nm excitation lines, along with specific filter sets for Alexa 488 (Zeiss filter set 38 HE) and Alexa 594 (Zeiss filter set 54 HE), was employed for visualizing and capturing images of fluorescent neurons. Images were taken at magnifications of ×10 to confirm placements of infusions in the DMS, ×20 and ×40 at five anterior–posterior levels (frontal cortex: 3.0 and 3.24 mm, striatum: 1.2, 0.5, and 0.2 mm) for verification and documentation of single- and double-labeled cells in cortex and striatum.

We employed a modification of a prior, semi-quantitative estimation of transfection efficacy and space (*Avila et al., 2020*). Briefly, GFP labeling that was centered in the striatal target region of prelimbic projections (*Mailly et al., 2013*) was assigned the highest transfection score (5), whereas labeling that extended into more lateral and ventral regions and only partially covered the target region were assigned lower scores. Likewise, mCherry labeling that exclusively covered the entire prelimbic cortex was assigned the highest score. Partial expression of mCherry in this region, or labeling that extended dorsally or ventrally beyond the prelimbic cortex, were assigned lower transfection efficacy

scores. Furthermore, single-and double-labeled neurons were counted in the prelimbic region, and the proportion of double-labeled neurons was computed for each rat. Counting frames were super-imposed on each subregion using the ImageJ multipoint tool. Cells were counted on two sections per brain, yielding four counts per brain and subregion. Averaged counts obtained from each brain and subregion were used for further analyses. To produce images of neurons expressing mCherry and GFP, and control for potential 'bleed-through' (e.g., *North, 2006*), we used the split view feature within Zen Black software (ZEN 2.31 SP1 Black Edition; Zeiss) for examination of individual channels with multi-track images. Tile scanning/stitching techniques were implemented at both ×10 (3 × 2 tiles, covering an area of 3455.94 by 2370.64 µm) and ×20 magnifications (4 × 3 tiles, covering an area of 1407.88 by 1135.31 µm).

## Experimental design and statistical analyses

### Behavioral data

Chi-square tests were used to determine the effects of sex and the commercial source of rats (vendor) on the distribution of PCA scores. Baseline CTTT performance was analyzed using conventional repeated measures ANOVAs; associated graphs depict individual values, means, and 95% confidence intervals (CIs). Conventional ANOVAs were computed using SPSS for Windows (version 28.0; SPSS) and GraphPad Prism (version 10.1.2).

### Glutamate concentrations and effects of CNO

Extracellular glutamate concentrations were sentinel-corrected, normalized to dopamine (if applicable), and pre-cue baseline corrected (see above for details). From these data, peak concentrations and the number of peaks were extracted (see above for the definition peaks). Given the complexity of the experimental design (variable number of rats per phenotype, turn and stop trials, and of cue-evoked glutamatergic transients), here we first determined potential phenotype-specific differences in glutamate peaks across trial types and response categories, followed, in cases, by post hoc analyses of event-related glutamate levels within individual phenotypes. LMMs with restricted maximum like-lihood estimation were computed in SPSS for Windows (version 28.0; SPSS). A key reason for using LMMs concerned the definition of sample size (number of animals vs. the number of individual traces collected from each animal). Moreover, sample sizes may vary across repeated or otherwise dependent measures (e.g., *Schielzeth et al., 2020*; *Yu et al., 2022*). Data from repeated sessions were used to compute repeated fixed effects, and phenotype was used to determine between-subjects fixed effects, with a subject identifier random intercept. Separate LMMs were computed for the analysis of glutamate levels during cue periods followed by turns, missed turns, or cued stops, and locked to reward delivery. The covariance structures with the lowest Akaike's information criterion were selected for each model (*Verbeke and Molenberghs, 2009*). Main effects of phenotype were followed up using Bonferroni's method for pairwise comparison of the means. When LMMs reported significant interactions, repeated measures ANOVAs were used to compare data from repeated sessions, with Huynh–Feldt-corrected $F$ values and corrected degrees of freedom applied in case of violation of the sphericity assumption. Data graphs showing LMM-analyzed glutamate concentrations depict individual values, estimated marginal means, and 95% CIs. Phenotypic frequencies of traces with one, two, or three peaks during cue periods were analyzed using Chi-squared tests.

### Contingency table analyses

Glutamate trace characteristics were extracted from 548 recordings of turn cue trials, 364 of which yielded a turn (GTs: 206, STs: 158) and 184 a miss (GTs: 112, STs: 72). Contingent on threshold maximum peak concentrations, and the presence of a single or multiple cue-evoked glutamate peaks, the proportions of turns and misses were compared using contingency table analyses (p values were computed using Fisher's exact test; GraphPad Prism). Furthermore, the relative probability for turns in GTs, as well as the reciprocal value, were derived from these analyses (Koopman asymptotic score; *Koopman, 1984*; *Motulsky, 2018*).

## Effects of CNO on CTTT performance

The effects of CNO or its vehicle on cued turns and stops, in rats expressing the inhibitory DREADD or the empty control vector, were analyzed using repeated measures ANOVA (general linear model) on the effects of phenotype and treatment day. For clarity, effects on the two behavioral measures were analyzed separately, at alpha = 0.05/2. Tukey's Honest Significant Difference test was used to compare, post hoc, the effects of the first and second administration of CNO with the effects of the first and second administration of vehicle (GraphPad Prism).

## p values and effect sizes

Exact p values were reported (*Greenwald et al., 1996*; *Sarter and Fritschy, 2008*; *Michel et al., 2020*). For key findings, effect sizes were computed using generalized eta squared ($\eta_p^2$) (effect sizes of 0.02 typically are classified as small, 0.13 as medium, and 0.26 as large; e.g., *Bakeman, 2005*), derived from Bonferroni pairwise comparisons of estimated marginal means (*Greenwald et al., 1996*).

## Acknowledgements

We thank Dr. Emma Reznick (University of Michigan) for assistance with the analysis of amperometry data and Dr. Kent Berridge (University of Michigan) for comments on a draft of the manuscript. The research described in this manuscript was supported by PHS grants R01DA045063 (MS) and P50NS091856 (Morris K Udall Center for 21 Excellence in Parkinson's Disease Research).

## Additional information

### Funding

| Funder | Grant reference number | Author |
| --- | --- | --- |
| National Institute on Drug Abuse | R01DA045063 | Martin Sarter |
| National Institute of Neurological Disorders and Stroke | P50NS091856 | Martin Sarter |

The funders had no role in study design, data collection and interpretation, or the decision to submit the work for publication.

### Author contributions

Cassandra Avila, Data curation, Software, Formal analysis, Investigation, Methodology, Writing – original draft, Writing – review and editing; Martin Sarter, Conceptualization, Supervision, Funding acquisition, Visualization, Writing – original draft, Project administration, Writing – review and editing

### Author ORCIDs

Martin Sarter ⬥ https://orcid.org/0000-0003-0441-9936

### Ethics

ll experimental procedures were approved by the University Committee on the Use and Care of Animals at the University of Michigan (protocol # PRO00010749 and PRO00011037) and carried out in laboratories accredited by the Association for Assessment and Accreditation of Laboratory Animal Care (AAALAC); Unit # 000285; PHS assurance # D16-00072 (A3114-01).

Reviewer #1 (Public review): https://doi.org/10.7554/eLife.100988.3.sa1
Reviewer #2 (Public review): https://doi.org/10.7554/eLife.100988.3.sa2
Reviewer #3 (Public review): https://doi.org/10.7554/eLife.100988.3.sa3
Author response https://doi.org/10.7554/eLife.100988.3.sa4

## Additional files

### Supplementary files
MDAR checklist

### Data availability
The datasets generated and analyzed for this study are publicly available in Dryad at https://doi.org/10.5061/dryad.8w9ghx3z1.

The following dataset was generated:

| Author(s) | Year | Dataset title | Dataset URL | Database and Identifier |
|---|---|---|---|---|
| Avila C, Sarter M | 2025 | Cortico-striatal action control inherent of opponent cognitive-motivational styles | https://doi.org/10.5061/dryad.8w9ghx3z1 | Dryad Digital Repository, 10.5061/dryad.8w9ghx3z1 |

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
