## [Editor Report · eLife Assessment]

This **valuable** manuscript investigated the role of glutamate signaling in the dorsomedial striatum of rats in a treadmill-based task and reported that it differs in goal-trackers compared to sign-trackers in a way that corresponds to differences in behaviour. The evidence supporting these claims is **solid** but could be further strengthened by adding more analyses and more detailed descriptions of current analyses. These findings will primarily be of interest to behavioural neuroscientists.

---

## [Referee Report · Reviewer #1 (Public review)]

Summary:

The authors measured glutamate transients in the DMS of rats as they performed an action selection task. They identified diverse patterns of behavior and glutamate dynamics depending on the pre-existing behavioral phenotype of the rat (sign tracker or goal tracker). Using pathway-specific DREADDs, they showed that these behavioral phenotypes and their corresponding glutamate transients were differentially dependent on input from the prelimbic cortex to the DMS.

Strengths:

Overall there are some very interesting results that make an important contribution to the field. Notably, the results seem to point to differential recruitment of the PL-DMS pathway in goal-tracking vs sign-tracking behaviors.

Weaknesses:

(1) The controls for off-target effects of CNO are not given sufficient importance both in terms of power and in reporting of their results. There is precedent to accept that CNO at the dosage given is unlikely to disrupt the behaviour, this doesn't justify the assumption that glutamate transmission won't be affected, and this possibility hasn't been sufficiently ruled out.

(2) The specificity of the viral approach needs to be clarified. Figure 8 indicates a large proportion of the PL neuron population that expresses mCherry in the absence of AAV-Cre. This infers that there are a large number of neurons inhibited by CNO administration that were outside the projection pathway, drawing into question the specificity of the effects.

---

## [Referee Report · Reviewer #2 (Public review)]

Summary:

The authors aimed to determine whether goal-directed and cue-driven attentional strategies (goal- and sign-tracking phenotypes) were associated with variation in cued motor responses and dorsomedial striatal (DMS) glutamate transmission. They used a treadmill task in which cues indicated whether rats should turn or stop to receive a reward. They collected and analyzed several behavioral measures related to task performance with a focus on turns (performance, latency, duration) for which there are more measures than for stops. First, they established that goal-trackers perform better than sign-trackers in post-criterion turn performance (cued turns completed) and turn initiation. They used glutamate sensors to measure glutamate transmission in DMS. They performed analyses on glutamate traces that suggest phasic glutamate DMS dynamics to cues were primarily associated with successful turn performance and were more characteristic of goal-trackers (ie. rats with "goal-directed" attentional strategy). Smaller and more frequent DMS glutamate peaks were associated with other task events, cued misses (missed turns), cued stops, and reward delivery and were more characteristic of sign-trackers (i.e. rats with "cue-driven" attentional strategies). Consistent with the reported glutamate findings, chemogenetic inhibition of prelimbic-DMS glutamate transmission had an effect on goal-trackers' turn performance without affecting sign-trackers' performance in the treadmill task.

Strengths:

The power of the sign- and goal-tracking model to account for neurobiological and behavioral variability is critically important to the field's understanding of heterogeneity of the brain in health and disease. The approach and methodology are sound in their contribution to this important effort.

The authors establish behavioral differences, measure a neurobiological correlate of relevance, and then manipulate that correlate in a broader circuitry and show a causal role in behavior that is consistent with neurobiological measurements and phenotypic differences.

Sophisticated analyses provide a compelling description of the authors' observations.

Limitations:

Considerable transparency was added in the revised preprint. The "n" for each analysis is now available in Tables 1 and 3, carefully cross-referenced by figure. Readers may now carefully consider the n's in drawing their own conclusions from reported data.

While more conventional trial-averaged population activity traces are not presented or analyzed, the unique nature of the peak phenotypes is likely to "wash out" potentially meaningful signals if averaged across subjects. The distribution of peaks analyses (and shifts observed with chemogenetic inhibition) are improved in the revised preprint and are informative to illustrate this likelihood. Representative traces should theoretically be consistent with population averages within phenotype, and if not, discussion of such inconsistencies may have enriched the conclusions drawn from the study. For example, population traces of the phasic cue response in GT may resemble the representative peak examples, while smaller irregular peaks of ST may "wash out" in a population average (possibly resulting in a prolonged elevation) and could have strengthened the rationale for more sophisticated analyses of peak probability that remain the focus of the revised preprint.

---

## [Referee Report · Reviewer #3 (Public review)]

Summary:

Avila and colleagues investigate the role of glutamate signaling in the dorsomedial striatum in a treadmill-based task where rats learn to turn or stop their walking based on learning cue-associations that allow them to acquire rewards. Phenotypic variation in Pavlovian conditioned sign and goal-tracking behavior was examined, where behavioral differences in stopping and turning were observed. Glutamate signals in the DMS were recording during the treadmill task, and were related to features of cue-controlled movement, with a stronger relationship seen for goal trackers. Finally, chemogenic inhibition of prelimbic neurons projecting to the DMS (the predicted source of those glutamate signals), preferentially affected cued movement in goal trackers. The authors couch these experiments in the context of cognitive control-attentional mechanisms, movement disorders, and individual differences in cue reactivity.

Strengths:

Overall these studies are interesting and are of general relevance to a number of research questions in neurology and psychiatry. The assessment of intersection of individual differences in cue-related learning strategies with movement-related questions - in this case cued turning behavior - is interesting and understudied question. The link between this work and growing notions of corticostriatal control of action selection makes it timely.

Weaknesses:

The clarity of the manuscript could be improved in several places, including in the graphical visualization of data. It is difficult to interpret the glutamate results, as presented, in the context of specific behaviors. It is difficult to assess how many trials/subjects are represented in the data shown, and too much emphasis is placed on representative examples. Averages traces of the glutamate data and other standard analysis approaches would improve the paper and allow for easier interpretation of the data.

---

## [Author Response]

The following is the authors’ response to the original reviews.

**Public Reviews:**

**Reviewer #1 (Public Review):**
Strengths:Overall there are some very interesting results that make an important contribution to the field. Notably, the results seem to point to differential recruitment of the PL-DMS pathway in goal-tracking vs sign-tracking behaviors.

Thank you.

Weaknesses:There is a lot of missing information and data that should be reported/presented to allow a complete understanding of the findings and what was done. The writing of the manuscript was mostly quite clear, however, there are some specific leaps in logic that require more elaboration, and the focus at the start and end on cholinergic neurons and Parkinson's disease are, at the moment, confusing and require more justification.

In the revised paper, we provide additional graphs and information in support of results, and we further clarify procedures and findings. Furthermore, we expanded the description of the proposed interpretational framework that suggests that the contrasts between the cortical-striatal processing of movement cues in sign- versus goal trackers are related to previously established contrasts between the capacity for the cortical cholinergic detection of attention-demanding cues.

**Reviewer #2 (Public review):**
Strengths:The power of the sign- and goal-tracking model to account for neurobiological and behavioral variability is critically important to the field's understanding of the heterogeneity of the brain in health and disease. The approach and methodology are sound in their contribution to this important effort.The authors establish behavioral differences, measure a neurobiological correlate of relevance, and then manipulate that correlate in a broader circuitry and show a causal role in behavior that is consistent with neurobiological measurements and phenotypic differences.Sophisticated analyses provide a compelling description of the authors' observations.

Thank you.

Weaknesses:It is challenging to assess what is considered the "n" in each analysis (trial, session, rat, trace (averaged across a session or single trial)). Representative glutamate traces (n = 5 traces (out of hundreds of recorded traces)) are used to illustrate a central finding, while more conventional trial-averaged population activity traces are not presented or analyzed. The latter would provide much-needed support for the reported findings and conclusions. Digging deeper into the methods, results, and figure legends, provides some answers to the reader, but much can be done to clarify what each data point represents and, in particular, how each rat contributes to a reported finding (ie. single trial-averaged trace per session for multiple sessions, or dozens of single traces across multiple sessions).Representative traces should in theory be consistent with population averages within phenotype, and if not, discussion of such inconsistencies would enrich the conclusions drawn from the study. In particular, population traces of the phasic cue response in GT may resemble the representative peak examples, while smaller irregular peaks of ST may be missed in a population average (averaged prolonged elevation) and could serve as a rationale for more sophisticated analyses of peak probability presented subsequently.

We have added two new Tables to clarify the number of rats per phenotype and sex used for each experiment described in the paper (Table 1), and the number of glutamate traces (range, median and total number) extracted for each analysis of performance-associated glutamate levels and the impact of CNO-mediated inhibition of fronto-striatal glutamate (Table 3).

As the timing of glutamate peaks varies between individual traces and subjects, relative to turn and stop cue onset or reward delivery, subject-and trial-averaged glutamate traces would “wash-out” the essential findings of phenotype- and task event-dependent patterns of glutamate peaks. In the detailed responses to the reviewers, we illustrate the results of an analysis of averaged traces to substantiate this view. Furthermore, as detailed in the section on statistical methods, and as mentioned by the reviewer under Strengths, we used advanced statistical methods to assure that data from individual animals contribute equally to the overall result, and to minimize the possibility that an inordinate number of trials obtained from just one or a couple of rats biased the overall analysis.

**Reviewer #3 (Public review):**
Strengths:Overall these studies are interesting and are of general relevance to a number of research questions in neurology and psychiatry. The assessment of the intersection of individual differences in cue-related learning strategies with movement-related questions - in this case, cued turning behavior - is an interesting and understudied question. The link between this work and growing notions of corticostriatal control of action selection makes it timely.

Thank you.

Weaknesses:The clarity of the manuscript could be improved in several places, including in the graphical visualization of data. It is sometimes difficult to interpret the glutamate results, as presented, in the context of specific behavior, for example.

We appreciate the reviewer’s concerns about the complexity of some of the graphics, particularly the results from the arguably innovative analysis illustrated in Figure 6. Figure 6 illustrates that the likelihood of a cued turn can be predicted based on single and combined glutamate peak characteristics. The revised legend for this figure provides additional information and examples to ease the readers’ access to this figure. In addition, as already mentioned above, we have added several graphs to further illustrate our findings.

**(Recommendations for the authors)**

**Reviewer #1 (Recommendations for the authors):**
(1) The differences in behavioral phenotype according to vendor (Figure 1c) are slightly concerning, could the authors please elaborate on why they believe this difference is? Are there any other differences in these stocks- i.e. weight, appearance, other types of behaviors?

Differences in PCA behavior across vendors or specific breeding colonies were documented previously and may reflect the impact of environmental, developmental and genetic factors (references added in the revised manuscript). We included animals from both vendors to increase phenotypic variability and due to animal procurement constraints during COVID-related restrictions.

(2) Possibly related to the above, the rats in Figure 1a and Figure 2 are different strains. Please clarify.

In the revised legend of Figure 2 we clarify that the rat shown in the photographs is a Long-Evans rat that was not part of the experiments described in this paper. This rat was used to generate these photos as the black-spotted fur provided better contrast against the white treadmill belt.

(3) Figure 3c, the pairwise comparison showing a significant increase from Day 1 to Day 3 is hard to understand unless this is a lasting change. Is this increase preserved at Day 4? Examination of either a linear trend across days or a simple comparison of either Day 1 & 2 against Day 3 & 4 or, minimally Day 1 against Day 4 would communicate this message. Otherwise, there doesn't seem to be much of a case for improvement across test sessions, which would also be fine in my view.

As the analysis of post-criterion performance also revealed an effect of DAY, we felt compelled to report and illustrate the results of pairwise comparisons in Fig. 3c. In agreement with the reviewer’s point, we did not further comment on this finding in the manuscript.

(4) Figure 4e. I find it extremely unlikely that every included electrode was located exactly at anterior 0.5mm. Please indicate the range - most anterior and most posterior of the included electrodes in the study.

The schematic section shown in Fig. 4e depicted that AP level of that section and collapsed all placements onto that level. As detailed in Methods, electrode placements needed to be within the following stereotaxic space: AP: -0.3 to 0.6 mm, ML: 2 to 2.5 mm, and DV: -4.2 to -5 mm (see Methods). To clarify this issue, the text in Results and the legend was modified and the 0.5 mm label was removed from Fig. 4e.

(5) The paper generally is quite data light and there are a lot of extra results reported that aren't shown in the figures. There are 17 instances of the phrase "not shown", some are certainly justified, but a lot of results are missing…

We followed the reviewer’s suggestion and added several graphs. The revised Figure 5 includes the new graph 5d that shows the number of glutamate traces with just 1, 2 or 3 peaks occurring during cue presentation period. Likewise, the revised Figure 7 includes the new graph 7h that shows the number of glutamate traces with just 1, 2 or 3 peaks following the administration of CNO or its vehicle. In both cases, we also revised the analysis of peak number data, by counting the number of cases (or traces) with just 1, 2 or 3 peaks and using Chi-squared tests to determine the impact of phenotype and, in the latter case, of CNO. In addition, the revised Figure 7 now includes a graph showing the main effects of phenotype and CNO in reward delivery-locked glutamate maximum peak concentrations (Fig. 7k). In revising these sections, we also removed the prior statement about glutamate current rise times as this isolated observation had no impact on subsequent analyses or the discussion.

Concerning the reviewer’s point 5d (*DMS eGFP transfection correlations* Figure 8), the manuscript clarifies that the absence of such a correlation was expected given that eGFP expression in the DMS does not accurately reproduce the prelimbic-DMS projection space that was inhibited by CNO. In contrast, the correlations between the efficacy of CNO and DREADD expression measures in prelimbic cortex were significant and are graphed (Figs. 8g and 8j).

(6) Please clarify the exact number of animals in each experiment. The caption of Figure 3 seems to suggest there are 29 GTs and 22 STs in the initial experiment, but the caption of Figure 5b seems to suggest there are N=30 total rats being analyzed (leaving 21 un-accounted for), or is this just the number of GTs (meaning there is one extra)?

We have added Table 1 to clarify the number of animals used across different experiments and stages. Additionally, we have included a new Table 3 that identifies, for each graph showing results from the analyses of glutamate concentrations, the number of rats from which recordings were obtained and the number of traces per rat (range, median, and total).

(7) Relatedly, in Figures 5c-f and Figures 7g-i, the data seem to be analyzed by trial rather than subject-averaged, please clarify and what is the justification for this?

As detailed Experimental design and statistical analyses, we employed linear mixed-effects modeling to analyze the amperometric data that generated figures 5 and 7 to minimize the risk of bias due to an excessive number of trials obtained from specific rats. LMMs were chosen to analyze these repeated (non-independent) data to address issues that may be present with subject-averaged data. For clarity, throughout the results for these figures, the numerator in the F-ratio reflects the degrees of freedom from the fixed effects (phenotype/sex) and the denominator reflects the error term influenced by the number of subjects and the within-subject variance.

Concerning the illustration and analysis of trial- or subject-averaged glutamate traces please see reviewer 2, point 1 and the graph in that section. Within a response bin, such as the 2-s period following turn cues, glutamate peaks – as defined in Methods - occur at variable times relative to cue onset. Averaging traces over a population of rats or trials would “wash-out” the phenotype- and task event-dependent patterns of glutamate concentration peaks, yielding, for example, a single, nearly 2-s long plateau for cue-locked glutamate recordings from STs (see Figure 5b versus the graph shown in response to reviewer 2, point 1).

(8) Likewise on page 22, the number of animals from which these trials were taken should be stated "The characteristics of glutamate traces (maximum peak concentration, number of peaks, and time to peak) were extracted from 548 recordings of turn cue trials, 364 of which yielded a turn (GTs: 206, STs: 158) and 184 a miss (GTs: 112, STs: 72).".

The number of animals is now included in the text and listed in Table 3.

(9) The control group for Figure 7 given the mCherry fluorophore - given the known off-target effects of CNO, this is a very important control. Minimally, this data should be shown, but it is troubling that the ST group has n=2, I don't really understand how any sort of sensible stats can be conducted with a group this size, and obviously it's too small to find any significant differences if they were there.

As discussed on p. 14-15 in the manuscript under the section *Clozapine N-Oxide*, the conversion rate of CNO to clozapine suggests that approximately 50-100 times the dose of clozapine (compared to our 5.0 mg/kg CNO dosage) would be required to produce effects on rodent behavior (references on p. 14-15).

Regarding evidence from control rats expressing the empty construct, the revised manuscript clarifies that no effects of CNO on cued turns were found in 5 GTs expressing the empty control vector. Although CNO had no effects in STs expressing the DREADD, we also tested the effects of CNO in 2 STs expressing the empty control vector (individual turn rates following vehicle and CNO are reported for these 2 STs). Moreover, we extracted turn cue-locked glutamate traces (vehicle: 18 traces; 16 CNO traces) from an empty vector-expressing GT and found that administration of CNO neither reduced maximum glutamate peak concentrations nor the proportion of traces with just one peak. The absence of effects of CNO on cued turning performance and on turn-cue locked glutamate dynamics are consistent with prior studies showing no effects of 5.0 mg/kg CNO in rats not expressing the DREADD vector (references in manuscript).

(10) Figure 8b - the green circle indicated by 1 is definitely not the DMS, this is the DLS, and animals with virus placement in this region should be excluded.

The reviewer of course is correct and that exactly was the point of that illustration, as such a transfection space would have received the lowest possible rating (as indicated by the “1” in the green space). Fig. 8b was intended to illustrate expression efficacy ratings and does not indicate actual viral transfection spaces. Because the results described in the manuscript did not include data from a brain with a striatal transfection space as was illustrated in green in the original Fig. 8b, we removed that illustration of an off-target transfection space.

(11) Figure 8j, the correlation specifically counts double-labeled PL hM4Di + eGFP neurons. Separating dual-labeled cells from all mCherry-labeled cells seems very strange given the nature of the viral approach. There seems to be an assumption that there are some neurons that express the mCherry-hM4Di that don't also have the AAV-Cre (eGFP). Obviously, if that were true this poses a huge problem for your viral approach and would mean that you're inhibiting a non-selective population of neurons. More likely, the AAV-Cre (eGFP) is present in all of your mCherry-hM4Di cells, just not at levels visible without GFP antibody amplification. Ideally, staining should be done to show that all cells with mCherry also have eGFP, but minimally this correlation should include all cells expressing mCherry with the assumption that they must also have the AAV-Cre.

As noted on page 15 in the Visualization and Quantification of eGFP/mCherry-Expressing Neurons section, eGFP expression in our viral approach was notably bright and did not necessitate signal enhancement. Furthermore, given the topographic organization of prelimbic-DMS projections on the on hand, and the variable transfection spaces in cortex and striatum on the other hand, the speculation that AAV-Cre may have been present in all mCherry cells is without basis. Second, there certainly are mCherry-positive cells that do not also express the retrogradely transported AAV-Cre, and that therefore were not affected by CNO. Third, the entire point of this dual vector strategy was to selectively inhibit prelimbic-striatal projections, and the strong correlation between double-labeled neuron numbers and cued turn scores substantiates the usefulness of this approach.

(12) Discussion, a bit more interpretation of the results would be good. Specifically - does the PL-DMS inhibition convert GTs to STs? There were several instances where the behavior and glutamate signals seemed to be pushed to look like STs but also a lot of missing data so it is hard to say. One would assume this kind of thing if, as I think is being said (please clarify), the ST phenotype is being driven by glutamatergic drive either locally or from sources other than PL cell bodies, presumably silencing the PL cell body inputs in GTs also leaves other glutamatergic inputs as the primary sources?

We agree with the reviewer that one could say, perhaps somewhat colloquially, that PL-DMS inhibition turns GTs to STs, in terms of turning performance and associated glutamate peak dynamics. The newly added data graphs are consistent with this notion. However, there are of course numerous other neurobiological characteristics which differ between GTs and STs and are revealed in the context of other behavioral or physiological functions. In the Discussion, and as noted by the reviewer, we discuss alternative sources of glutamatergic control in STs and the functional implications of bottom-up mechanisms. In the revised manuscript, we have updated references and made minor revisions to improve this perspective.

(13) I found the abstract really detailed and very dense, it is pretty hard to understand in its current form for someone who hasn't yet read the paper. At this level, I would recommend more emphasis on what the results mean rather than listing the specific findings, given that the task is still quite opaque to the reader.

We revised the abstract, in part by deleting two rather dense but non-essential statements of results and by adding a more accessible conclusion statement.

(14) There are a lot of abbreviations: CTTT, PD, PCA, GT, ST, MEA, GO, LMM, EMMs, PL, DMS. Some of these are only mentioned a few times: MEA, LMM, and EMMs are all mentioned less than 5 times. To reduce mental load for the reader, you could spell these ones out, or include a table somewhere with all of the abbreviations.

We added a list of Abbreviations and Acronyms and eliminated abbreviations that were used infrequently.

(15) Generally, the logic that cortico-striatal connections contribute to GT vs ST seems easy to justify, however, the provided justification is missing a line of connection: "As such biases of GTs and STs were previously shown to be mediated in part via contrasting cholinergic capacities for the detection of cues (Paolone et al., 2013; Koshy Cherian et al., 2017; Pitchers et al., 2017a; Pitchers et al., 2017b), we hypothesized that contrasts in the cortico-striatal processing of movement cues contribute to the expression of these opponent biases." Please elaborate on why specifically cholinergic involvement suggests corticostriatal involvement. I think there are probably more direct reasons for the current hypothesis.

Done – see p. 4-5.

(16) Along the same line, paragraph 3 of the intro about Parkinson's disease and cholinergics seems slightly out of place. This is because the specific or hypothesized link between these things and corticostriatal glutamate has not been made clear. Consider streamlining the message specifically to corticostriatal projections in the context of the function you are investigating.

Done – see p. 4-5.

(17) Page 8, paragraph 2. There is a heading or preceding sentence missing from the start of this paragraph: "Contrary to the acclimation training phase, during which experimenters manually controlled the treadmill, this phase was controlled entirely by custom scripts using Med-PC software and interface (MedAssociates).".

Revised and clarified.

(18) Page 13 "We utilized a pathway-specific dual-vector chemogenetic strategy (e.g., Sherafat et al., 2020) to selectively inhibit the activity of fronto-cortical projections to the DMS". The Hart et al (2018) reference seems more appropriate being both the same pathway and viral combination approach.

Yes, thank you, we’ve updated the citation.

(19) Pages 20-21: "Maximum glutamate peak concentrations recorded during the cue period were significantly higher in GTs than in STs (phenotype: F(1,28.85) = 8.85, P=0.006, ηp 2=0.23; Fig. 5c). In contrast, maximum peak amplitudes locked to other task events all were significantly higher in STs." The wording here is misleading, both Figures 5c and 5d report glutamate peaks during the turn cue, the difference is what the animal does. So, it should be something like "Maximum glutamate peak concentrations recorded during the cue period were significantly higher in GTs than in STs when the animal correctly made a turn (stats) but this pattern reversed on missed trials when the animal failed to turn (stats)..." or something similar.

Yes, thank you. We have revised this section accordingly.

(20) Same paragraph: "Contingency tables were used to compare phenotype and outcome-specific proportions and to compute the probability for turns in GTs relative to STs." What is an outcome-specific proportion?

This has been clarified.

.

(21) Page 22 typo: "GTs were only 0.74 times as likely as GTs to turn".

Fixed.

(22) The hypothesis for the DREADDs experiment isn't made clear enough. Page 23 "In contrast, in STs, more slowly rising, multiple glutamate release events, as well as the presence of relatively greater reward delivery-locked glutamate release, may have reflected the impact of intra-striatal circuitry and ascending, including dopaminergic, inputs on the excitability of glutamatergic terminals of corticostriatal projections" As far as I can understand, the claim seems to be that glutamate release might be locally modulated in the case of ST, on account of the profile of glutamate release- more slowly rising, multiple events, and reward-locked. Please clarify why these properties would preferentially suggest local modulation.

We have revised and expanded this section to clarify the basis for this hypothesis.

(23) The subheadings for the section related to Figure 7 "CNO disrupts..." "CNO attenuates..." presumably you mean fronto-striatal inhibition disrupts/attenuates. As it stands, it reads like the CNO per se is having these effects, off-target.

Fixed.

(24) The comparison of the results in the discussion against a "hypothetical" results section had the animals not been phenotyped behaviorally is unnecessary and overly speculative, given that 30-40% of rats don't fall into either of these two categories. I think the point here is to emphasize the importance of taking phenotype into account. This point can surely be made directly in its own sentence, probably somewhere towards the end of the discussion.

We have partly followed the reviewer’s advice and separated the discussion of the hypothetical results from the summary of main findings. However, we did not move this discussion toward the end of the Discussion section as we believe that it justifies the guiding focus of the discussion on the impact of phenotype.

(25) The discussion, like the introduction, talks a lot about cholinergic activity. As noted, this link is unclear - particularly how it links with the present results, please clarify or remove. Likewise high-frequency oscillations.

We have revised relevant sections in the Introduction (see above) and Discussion sections. However, given the considerable literature indicating contrasts between the cortical cholinergic-attentional capacities of GTs and STs, the interpretation of the current findings in that larger context is justified.

(26) Typo DSM in the discussion x 2.

Thanks, fixed.

**Reviewer #2 (Recommendations for the authors):**
(1) As mentioned in the Public Review, it is challenging to assess what is considered the "n" in each analysis, particularly for the glutamate signal analysis (trial, session, rat, trace (averaged across session or single trial)). Representative glutamate traces are used to illustrate a central finding, while more conventional trial-averaged population activity traces are not presented or analyzed. For example, n = 5 traces, out of hundreds of recorded traces, with each rat contributing 1-27 traces across multiple sessions suggests ~1-2% of the data are shown as time-resolved traces. Representative traces should in theory be consistent with population averages within phenotype, and if not, discussion of such inconsistencies would enrich the conclusions drawn from the study. In particular, population traces of the phasic cue response in GT may resemble the representative peak examples, while smaller irregular peaks of ST may be missed in a population average (averaged prolonged elevation in signal) and could serve as rationale for more sophisticated analyses of peak probability presented subsequently (and relevant to opening paragraph of discussion where hypothetical data rationale is presented).

We have added the new Table 1 to provide a complete account of the number of rats, per phenotype and sex, for each component of the experiments. In addition, the new Table 3 provides the range, median and total number of glutamate traces that were analyzed and formed the foundation of the individual data graphs depicting the results of glutamate concentration analyses.

We chose not to present trial- or subject-averaged traces, as glutamate peaks occur at variable times relative to the onset of turn and stop cues and reward delivery, and therefore averaging across a population of rats or trials would obscure phenotype- and task event-dependent patterns of glutamate peaks. The attached graph serves to illustrate this issue. The graph shows turn cue-locked glutamate concentrations (M, SD) from trials that yielded turns, averaged over all traces used for the analysis of the data shown in Fig. 5d (see also Table 3, top row). Because of the variability of peak times, trial- and subject-averaging of traces from STs yielded a nearly 2-s long elevated plateau of glutamate concentrations (red triangles), contrasting with the presence single and multiple peaks in STs as illustrated in Figs. 5b and 5e. Furthermore, averaging of traces from GTs obscured the presence of primarily single turn cue-locked peaks. Because of the relatively large variances of averaged data points, again reflecting the variability of peak times, analysis of glutamate levels during the cue period did not indicate an effect of phenotype (F(1,190)=1.65, *P*=0.16). Together, subject- or trial-averaged traces would not convey the glutamate dynamics that form the essence of the amperometric findings obtained from our study. We recognize, as inferred by the reviewer, that smaller irregular peaks in STs may have been missed given the definition of a glutamate peak (see Methods). It is in part for that reason that we conducted a prospective analysis of the probability for turns given a combination of peak characteristics (maximum peak concentration and peak numbers; Fig. 6).

(2)To this latter point, the relationship between the likelihood to turn and the size of glutamate peak is focused on the GT phenotype, which limits understanding of how smaller multiple peaks relate to variables of interest in ST (missed turns, stops, reward). If it were possible to determine the likelihood for each phenotype, without a direct contrast of one phenotype relative to the other, this would be a more straightforward description of how signal frequency and amplitude relate to relevant behaviors in each group. Depending on the results, this could be done in addition to or instead of the current analysis in Figure 6.

We considered the reviewer’s suggestion but could not see how attempts to analyze the role of maximum glutamate concentrations and number of peaks within a single phenotype would provide any significant insights beyond the current description of results. Moreover, as stressed in the 2nd paragraph of the Discussion (see Reviewer 1, point 24), the removal of the phenotype comparison would nearly completely abolish the relationships between glutamate dynamics and behavior from the current data set.

**Author response image 1. sa4fig1:** 

(3) If Figure 6 is kept, a point made in the text is that GT is 1.002x more likely than ST to turn at a given magnitude of Glu signal. 1.002 x more likely is easily (perhaps mistakenly) interpreted as nearly identical likelihood. Looking closely at the data, perhaps what is meant is @ >4uM the difference between top-line labeled {b} and bottom-line labeled {d,e} is 1.002? If not, there may be a better way to describe the difference as 1x could be interpreted as the same/similar.

Concerning the potential for misinterpretation, the original manuscript stated (key phrase marked here in red font): Comparing the relative turn probabilities at maximum peak concentrations >4 µM, GTs were 1.002 times more likely (or nearly exactly twice as likely) as STs to turn if the number of cue-evoked glutamate peaks was limited to one (rhombi in Fig. 6a) when compared to the presence of 2 or 3 peaks (triangles in Fig. 6a). However, we appreciate the reviewer’s concern about the complexity of this statement and, as it merely re-emphasized a result already described, it was deleted.

(4) For Figure 7e, the phenotype x day interaction is reported, but posthocs are looking within phenotype (GT) at treatment effects. Is there a phenotype x day x treatment, or simply phenotype x treatment (day collapsed) to justify within-group treatment posthocs?

We have revised the analysis and illustration of the data shown in Figs 7e and 7f, by averaging the test scores from the two tests, per animal, of the effects of vehicle and CNO, to be able to conduct a simpler 2-way analysis of the effects of phenotype and treatment.

(5) Ideally, viral control is included as a factor in this analysis as well. The separate analysis for viral controls was likely done due to low n, however negative findings from an ANOVA in which an n=2 (ST) should be interpreted with extreme caution. The authors already have treatment control (veh, CNO) and may consider dropping the viral controls completely due to the lack of power to perform appropriate analyses.

This issue has been clarified – see reviewer 1, point 9.

Minor:(1) In the task description, it could be clearer how reward delivery relates to turns and stops. For example, does the turn cue indicate the rat will be rewarded at the port behind it? Does the stop cue indicate that the rat will be rewarded at the port in front of it? This makes logical sense, but the current text does not describe the task in this way, instead focusing on what is the correct action (seemingly but unlikely independent of reinforcement).

We have updated the task description in Methods and the legend of Figure 2 to indicate the location of reward delivery following turns and stops.

(2) For the peak analysis, what is the bin size for determining peaks? It is indicated that the value before and after the peak is >1 SD below the peak value, so it is helpful to know the temporal bin resolution for this definition.

As detailed on p 11-12 under Amperometry Data Processing and Analysis of Glutamate Peaks, we analyzed glutamate concentrations recorded at a frequency of 5 Hz (200 ms bins) throughout the 2-second-long presentation of turn and stop cues and for a 2-second period following reward delivery.

(3) Long Evans rats are pictured in Figure 2 (presumably contrast with a white background is better here), while SD rats are pictured in Figure 1. Perhaps stating why LE rats are pictured would help clear up any ambiguity about the strains used, as a quick look gives the impression two strains are used in two different tasks.

Yes, see reviewer 1, point 2.

(4) In Figure 7e, the ST and GT difference in turns/turn cue does not seem to replicate prior findings for tracking differences for this measure (Figure 3b). ST from the chemogenetic cohort seems to perform better than rats whose behavior was examined prior to glutamate sensor insertion. What accounts for this difference? Training and testing conditions/parameters?

The reviewer is correct. The absence of a significant difference between vehicle-treated GTs and vehicle-treated STs in Fig. 7e reflects a relatively lower turn rate in GTs than was seen in the analysis of baseline behavior (Fig. 3b; note the different ordinates of the two figures, needed to show the impact of CNO in Fig. 7e). Notably, the data in Fig. 7e are based on fewer rats (12 versus 29 GTs and 10 versus 22 STs; Table 1) and on rats which at this point had undergone additional surgeries to infuse the DREADD construct and implant electrode arrays. We can only speculate that these surgeries had greater detrimental effects in GTs, perhaps consistent with evidence suggesting that immune challenges trigger a relatively greater activation of their innate immune system (Carmen et al., 2023). We acknowledged this issue in the revised Results.

(5) The authors are encouraged to revise for grammar (are vs. is, sentence ending with a preposition, "not only" clause standing alone) and word choice (i.e. in introduction: insert, import, auditorily). Consider revising the opening sentence on page 5 for clarity.

We have revised the entire text to improve grammar and word choice.

(6) Do PD fallers refer to rats or humans? if the latter, this may be a somewhat stigmatizing word choice.

We have replaced such phrases using more neutral descriptions, such as referring to people with PD who frequently experience falls.

(7) Page 27 What does "non-instrumental" behavior mean?

We have re-phrased this statement without using this term.

(8) The opening paragraph of the discussion is focused on comparing reported results (with phenotype as a factor) to a hypothetical description of results (without phenotype as a factor) that were not presented in the results section. There is one reference to a correlation analysis on collapsed data, but otherwise, no reporting of data overall rats without phenotype as a factor. If this is a main focus, including these analyses in the results would be warranted. If this is only a minor point leading to discussion, authors could consider omitting the hypothetical comparison.

We have revised this section - see reviewer 1 point 24.

**Reviewer #3 (Recommendations for the authors):**
(1) These are really interesting studies. I think there are issues in data presentation/analysis that make it difficult to parse what exactly is happening in the glutamate signals, and when. Overall the paper is just a bit of a difficult read. A generally standard approach for showing neural recording data of many kinds, including, for example, subject-averaged traces, peri-event histograms, heatmaps, etc summarizing and quantifying the results - would be helpful. Beyond the examples in Figure 5, I would suggest including averaged traces of the glutamate signals and quantification of those traces.

We have addressed these issues in multiple ways, see the response to several points of reviewers 1 and 2, particularly reviewer 2, point 1.

(2) Figure 6 (and the description in the response letter) is also very non-intuitive. It's unclear how the examples shown relate to the reported significance indicators/labels/colors etc in the figure. I would suggest rethinking this figure overall, and if there is a more direct quantitative way to connect signal features with behavior. Again, drawing from standard visualization approaches for neural data could be one approach.

See also reviewer 2 points 1 and 3. Furthermore, we have revised the text in Results and the legend to improve the accessibility of Fig. 6.

(3) As far as I can tell, all of the glutamate sensor conclusions reflect analysis collapsed across 100s of trials. Do any of the patterns hold for a subjects-wise analysis? How variable are individual subjects?

We employed linear mixed-effect model analyses and added a random subject intercept to account for subject variability outside fixed effects (phenotype and treatment). The variance of the intercept ranged 0.01-1.71 SEM across outcome (cued turns/cued stops/misses). See also reviewer 1, point 7 and reviewer 2, point 1.